# On the Curses of Future and History in Future-dependent Value Functions for OPE

**Yuheng Zhang**
University of Illinois Urbana-Champaign
yuhengz2@illinois.edu

**Nan Jiang**
University of Illinois Urbana-Champaign
nanjiang@illinois.edu

## Abstract

We study off-policy evaluation (OPE) in partially observable environments with complex observations, with the goal of developing estimators whose guarantee avoids exponential dependence on the horizon. While such estimators exist for MDPs and POMDPs can be converted to history-based MDPs, their estimation errors depend on the state-density ratio for MDPs which becomes history ratios after conversion, an exponential object. Recently, Uehara et al. [2022a] proposed *future-dependent value functions* as a promising framework to address this issue, where the guarantee for memoryless policies depends on the density ratio over the *latent* state space. However, it also depends on the boundedness of the future-dependent value function and other related quantities, which we show could be exponential-in-length and thus erasing the advantage of the method. In this paper, we discover novel coverage assumptions tailored to the structure of POMDPs, such as *outcome coverage* and *belief coverage*, which enable polynomial bounds on the aforementioned quantities. As a side product, our analyses also lead to the discovery of new algorithms with complementary properties.

## 1 Introduction and Related Works

Off-policy evaluation (OPE) is the problem of estimating the return of a new *evaluation policy* $\pi_e$ based on historical data, which is typically collected using a different policy $\pi_b$ (the *behavior policy*). OPE plays a central role in the pipeline of offline reinforcement learning (RL), but is also notoriously difficult. Among the major approaches, importance sampling (IS) and its variants Precup et al. [2000], Jiang and Li [2016] provide unbiased and/or asymptotically correct estimation using the *cumulative importance weights*: given a trajectory of observations and actions $o_1, a_1, o_2, a_2, \ldots, o_H, a_H$, the cumulative importance weight is $\prod_{h=1}^{H} \frac{\pi_e(a_h|o_h)}{\pi_b(a_h|o_h)}$, whose variance grows *exponentially* with the horizon, unless $\pi_b$ and $\pi_e$ are very close in their action distributions. In the language of offline RL theory [Chen and Jiang, 2019, Xie and Jiang, 2021, Yin and Wang, 2021], the boundedness of the cumulative importance weights is the coverage assumption required by IS, a very stringent one.

Alternatively, algorithms such as Fitted-Q Evaluation [FQE; Ernst et al., 2005, Munos and Szepesvári, 2008, Le et al., 2019] and Marginalized Importance Sampling [MIS; Liu et al., 2018, Xie et al., 2019, Nachum et al., 2019, Uehara et al., 2020] enjoy more favorable coverage assumptions, at the cost of function-approximation biases. Instead of requiring bounded cumulative importance weights, FQE and MIS only require that of the *state-density ratios*, which can be substantially smaller [Chen and Jiang, 2019, Xie and Jiang, 2021]. That is, when the environment satisfies the Markov assumption (namely $o_h$ is a *state*), the guarantees of FQE and MIS only depend on the range of $d_h^{\pi_e}(o_h)/d_h^{\pi_b}(o_h)$, where $d_h^{\pi_e}$ and $d_h^{\pi_b}$ is the marginal distribution of $o_h$ under $\pi_e$ and $\pi_b$, respectively.

In this paper, we study the non-Markov setting, which is ubiquitous in real-world applications. Such environments are typically modeled as Partially Observable Markov Decision Processes (POMDPs)

Kaelbling et al. [1998]. Despite the more general formulation, one can reduce a POMDP to an MDP, making algorithms for MDPs applicable: we can simply define an equivalent MDP, with its state being the *history* of the original POMDP, $(o_1, a_1, \ldots, o_h)$. Unfortunately, a close inspection reveals the problem: the state-density ratio after conversion is

$$\frac{d_h^{\pi_e}(o_1, a_1, \ldots, o_h)}{d_h^{\pi_b}(o_1, a_1, \ldots, o_h)} = \prod_{h'=1}^{h-1} \frac{\pi_e(a_{h'}|o_{h'})}{\pi_b(a_{h'}|o_{h'})},$$

which is exactly the cumulative importance weights in IS and thus also an exponential object!

To address this issue, Uehara et al. [2022a] recently proposed a promising framework called *future-dependent value functions* (or FDVF for short). Notably, their coverage assumption is the boundedness of density ratios between $\pi_e$ and $\pi_b$ over the *latent state* for memoryless policies. This is as if we were dealing directly with the latent MDP underlying the POMDP, a perhaps best possible scenario. Nevertheless, *have we achieved exponential-free OPE in POMDPs?*

The answer to this question turns out to be nontrivial. In addition to the latent-state coverage parameter, the guarantee in Uehara et al. [2022a] also depends on other quantities that are less interpretable. Among them, the boundedness of FDVF itself—a concept central to this framework—is unclear, and we show that a natural construction yields an upper bound that still scales with the cumulative importance weights, thus possibly erasing the superiority of the framework over IS or MDP-reduction.

In this work, we address these caveats by proposing novel coverage assumptions tailored to the structure of POMDPs, under which fully polynomial estimation guarantees can be established. More concretely, our contributions are:

1. For FDVFs, we show that a novel coverage concept called ***outcome coverage*** is sufficient for guaranteeing its boundedness (Section 4). Notably, outcome coverage concerns the overlap between $\pi_b$ and $\pi_e$ *from the current time step onward*, whereas all MDP coverage assumptions concern that *before* the current step.

2. With another novel concept called ***belief coverage*** (Section 5.1), we establish fully polynomial estimation guarantee for the algorithm in Uehara et al. [2022a]. The discovery of belief coverage also leads to a novel algorithm (Section 5.3) that is analogous to MIS for MDPs.

3. Despite the similarity to linear MDP coverage [Duan et al., 2020] due to the linear-algebraic structure, these POMDP coverage conditions also have their own unique properties due to the $L_1$ normalization of belief and outcome vectors. We present improved analyses that leverage such properties and avoid explicit dependence on the size of the latent state space (Section 4.2).

## 2 Preliminaries

**POMDP Setup.** We consider a finite-horizon POMDP $\left\langle H, \mathcal{S} = \bigcup_{h=1}^{H} \mathcal{S}_h, \mathcal{A}, \mathcal{O} = \bigcup_{h=1}^{H} \mathcal{O}_h, R, \mathbb{O}, \mathbb{T}, d_1 \right\rangle$, where $H$ is the horizon, $\mathcal{S}_h$ is the latent state space at step $h$ with $|\mathcal{S}_h| = S$, $\mathcal{A}$ is the action space with $|\mathcal{A}| = A$, $\mathcal{O}_h$ is the observation space at step $h$ with $|\mathcal{O}_h| = O$, $R : \mathcal{O} \times \mathcal{A} \to [0, 1]$ is the reward function, $\mathbb{O} : \mathcal{S} \to \Delta(\mathcal{O})$ is the emission dynamics with $\mathbb{O}(\cdot|s_h)$ supported on $\mathcal{O}_h$ for $s_h \in \mathcal{S}_h$, $\mathbb{T} : \mathcal{S} \times \mathcal{A} \to \Delta(\mathcal{S})$ is the dynamics ($\mathbb{T}(\cdot|s_h, a_h)$ is supported on $\mathcal{S}_{h+1}$), and $d_1 \in \Delta(\mathcal{S}_1)$ is the initial latent state distribution. For mathematical convenience we assume all the spaces are finite and discrete, but the cardinality of $\mathcal{O}_h$, $O$, **can be arbitrarily large**. A trajectory (or episode) is sampled as $s_1 \sim d_1$, then $o_h \sim \mathbb{O}(\cdot|s_h)$, $r_h = R(o_h, a_h)$, $s_{h+1} \sim \mathbb{T}(\cdot|s_h, a_h)$ for $1 \leq h \leq H$, with $a_{1:H}$ decided by the decision-making agent, and the episode terminates after $a_H$. $s_{1:H}$ are latent and not observable to the agent.

**History-future Split.** Given an episode $o_1, a_1, \ldots, o_H, a_H$ and a time step $h$ of interest, it will be convenient to rewrite the episode as

$$(\tau_h, \overbrace{o_h, a_h, f_{h+1}}^{f_h}).$$

Here $\tau_h = (o_1, a_1, \ldots o_{h-1}, a_{h-1}) \in \mathcal{H}_h := \prod_{h'=1}^{h-1}(\mathcal{O}_h \times \mathcal{A})$ denotes the historical observation-action sequence (or simply *history*) prior to step $h$, and $f_{h+1} = (o_{h+1}, a_{h+1}, \ldots, o_H, a_H) \in \mathcal{F}_{h+1} := \prod_{h'=h+1}^{H}(\mathcal{O}_{h'} \times \mathcal{A})$ denotes the *future* after step $h$. This format will be convenient for

reasoning about the system dynamics at step $h$. We use $\mathcal{H} = \bigcup_{h=1}^{H} \mathcal{H}_h$ to denote the entire history domain and use $\mathcal{F} = \bigcup_{h=1}^{H} \mathcal{F}_h$ to denote the entire future domain. This way we can use stationary notation for functions over $\mathcal{S}, \mathcal{O}, \mathcal{H}, \mathcal{F}$, where the time step can be identified from the function input (e.g., $R(o_h, a_h)$), and functions with time-step subscripts refer to their restriction to the $h$-th step input space, often treated as a vector (e.g., $R_h \in \mathbb{R}^{\mathcal{O}_h \times \mathcal{A}}$).

**Memoryless and History-dependent Policies.** A policy $\pi : \bigcup_{h=1}^{H}(\mathcal{H}_h \times \mathcal{O}_h) \to \Delta(\mathcal{A})$ specifies the action probability conditioned on the past observation-action sequence.[1] In the main text, we will restrict ourselves to *memory-less* (or reactive) policies that only depends on the current observation $o_h$; extension to general policies is similar to Uehara et al. [2022a] and discussed in Appendix B.6. For any $\pi$, we use $\Pr_\pi$ and $\mathbb{E}_\pi$ for the probabilities and expectations under episodes generated by $\pi$, and define $J(\pi)$ as the expected cumulative return: $J(\pi) := \mathbb{E}_\pi\left[\sum_{h=1}^{H} R(o_h, a_h)\right]$. For memoryless $\pi$, we also define $V_\mathcal{S}^\pi(s_h)$ as the latent state value function at $s_h$: $V_\mathcal{S}^\pi(s_h) := \mathbb{E}_\pi\left[\sum_{h'=h}^{H} R(o_{h'}, a_{h'}) \mid s_h\right] \in [0, H]$. $d^\pi(s_h)$ denotes the marginal distribution of $s_h$ under $\pi$.

**Off-policy Evaluation.** In OPE, the goal is to estimate $J(\pi_e)$ using $n$ data trajectories $\mathcal{D} = \{(o_1^{(i)}, a_1^{(i)}, r_1^{(i)}, \ldots, o_H^{(i)}, a_H^{(i)}, r_H^{(i)}) : i \in [n]\}$ collected using $\pi_b$. We write $\mathbb{E}_\mathcal{D}[\cdot]$ to denote empirical approximation of expectation using $\mathcal{D}$. Define the one-step action probability ratio $\mu(o_h, a_h) := \frac{\pi_e(a_h|o_h)}{\pi_b(a_h|o_h)}$. We make the following assumption throughout:

**Assumption 1** (Action coverage). We assume $\pi_b(a_h|o_h)$ is known and $\max_{h, o_h, a_h} \mu(o_h, a_h) \leq C_\mu$.

This is a standard assumption in the OPE literature, and is needed by IS and value-based estimators that model state value functions [Jiang and Li, 2016, Liu et al., 2018].

**Belief and Outcome Matrices.** We now introduce two matrices of central importance to our discussions. Given history $\tau_h$, we define $\mathbf{b}(\tau_h) \in \mathbb{R}^\mathcal{S}$ as its belief state vector where $\mathbf{b}_i(\tau_h) = \Pr(s_h = i|\tau_h)$. Then the **belief matrix** $M_{\mathcal{H},h} \in \mathbb{R}^{S \times \mathcal{H}_h}$ is one where the column indexed by $\tau_h \in \mathcal{H}_h$ is $\mathbf{b}(\tau_h)$. Similarly, for future $f_h$, we define $\mathbf{u}(f_h) \in \mathbb{R}^S$ as its outcome vector where $[\mathbf{u}(f_h)]_i = \Pr_{\pi_b}(f_h|s_h = i)$. The **outcome matrix** $M_{\mathcal{F},h} \in \mathbb{R}^{S \times \mathcal{F}_h}$ is one where the column indexed by $f_h$ is $\mathbf{u}(f_h)$. Unlike the belief matrix, the outcome matrix $M_{\mathcal{F},h}$ is dependent on the behavior policy $\pi_b$, which is omitted in the notation.

For mathematical conveniences, we make the following assumptions throughout merely for simplifying presentations; they allow us to invert certain covariance matrices and avoid $0/0$ situations, which can be easily handled with extra care when the assumptions do not hold.

**Assumption 2** (Invertibility). $\forall h \in [H]$, (1) $\mathrm{rank}(M_{\mathcal{H},h}) = \mathrm{rank}(M_{\mathcal{F},h}) = S$.
(2) $\forall f_h$, $\Pr_{\pi_b}(f_h) > 0$; $\forall o_h, a_h$, $R(o_h, a_h) > 0$.

**Other Notation.** Given a vector $\mathbf{a}$, $\|\mathbf{a}\|_\Sigma := \sqrt{\mathbf{a}^\top \Sigma \mathbf{a}}$, where $\Sigma$ is a positive semi-definite (PSD) matrix. When $\Sigma = \mathrm{diag}(d)$ for a stochastic vector $d$, this is the $d$-weighted 2-norm of $\mathbf{a}$, which we also write as $\|\mathbf{a}\|_{2,d}$. For a positive integer m, we use $[m]$ to denote the set $\{1, 2, \cdots, m\}$. For a matrix $M$, we use $(M)_{ij}$ to denote the $ij$ entry of $M$.

## 3 Future-dependent Value Functions

In this section, we provide a recap of FDVFs and translate the main result of Uehara et al. [2022a] into the finite-horizon setting, which is mathematically cleaner and more natural in many aspects; see Appendix B.2 for further discussion.

To illustrate the main idea behind FDVFs, recall that the tool that avoids the exponential weights in MDPs is to model the *value functions*. While we would like to apply the same idea to POMDPs,

---

[1]Here we do *not* allow the policy to depend on the latent state, which satisfies sequential ignorability and eliminates data confoundedness; see Namkoong et al. [Section 3; 2020] and Uehara et al. [Appendix B; 2022a]. There is also a line of research on OPE in confounded POMDPs where the behavior policy *only* depends on the latent state [Tennenholtz et al., 2020, Shi et al., 2022]; see Appendix A.

history-dependent value functions lead to unfavorable coverage conditions (see Section 1). The only other known notion of value functions we are left with is that over the latent state space, $V_{\mathcal{S}}^{\pi_e}(s_h)$, which unfortunately is not accessible to the learner since it operates on unobservable latent states.

The central idea is to find *observable proxies* of $V_{\mathcal{S}}^{\pi_e}(s_h)$, which takes *future* as inputs:

**Definition 3** (Future-dependent value functions [Uehara et al., 2022a]). A future-dependent value function $V_{\mathcal{F}} : \mathcal{F} \rightarrow \mathbb{R}$, where $\mathcal{F} := \bigcup_h \mathcal{F}_h$, is any function that satisfies the following: $\forall s_h$, $\mathbb{E}_{\pi_b}[V_{\mathcal{F}}(f_h) \mid s_h] = V_{\mathcal{S}}^{\pi_e}(s_h)$. Equivalently, in matrix form, we have $\forall h$,

$$M_{\mathcal{F},h} \times V_{\mathcal{F},h} = V_{\mathcal{S},h}^{\pi_e}. \tag{1}$$

Recall our convention, that $V_{\mathcal{F},h} \in \mathbb{R}^{|\mathcal{F}_h|}$ is $V_{\mathcal{F}}$ restricted to $\mathcal{F}_h$, and $V_{\mathcal{S},h}^{\pi_e}$ is defined similarly.

A FDVF $V_{\mathcal{F}}$ is a property of $\pi_e$, but also depends on $\pi_b$. As we will see later in Section 4, the boundedness of $V_{\mathcal{F}}$ will depend on certain notion of coverage of $\pi_b$ over $\pi_e$. As another important property, the FDVF $V_{\mathcal{F}}$ is generally not unique even if we fix $\pi_e$ and $\pi_b$, as Eq.(1) is generally an underdetermined linear system ($S \ll |\mathcal{F}_h|$) and can yield many solutions. As we see below, it suffices to model *any* one of the solutions. Thus, from now on, when we talk about the boundedness of $V_{\mathcal{F}}$, we always consider the $V_{\mathcal{F}}$ with the smallest range among all solutions.

**Finite Sample Learning.** Like in MDPs, to learn an approximate FDVF from data, we will minimize some form of estimated Bellman residuals (or errors). For that we need to first introduce the Bellman residual operators for FDVFs:

**Definition 4** (Bellman residual operators). $\forall V : \mathcal{F} \rightarrow \mathbb{R}$, the Bellman residual on state $s_h$ is:[2]

$$(\mathcal{B}^{\mathcal{S}}V)(s_h) := \mathbb{E}_{\substack{a_h \sim \pi_e \\ a_{h+1:H} \sim \pi_b}}[r_h + V(f_{h+1}) \mid s_h] - \mathbb{E}_{\pi_b}[V(f_h) \mid s_h].$$

Similarly, the Bellman residual onto the history $\tau_h$ is:

$$(\mathcal{B}^{\mathcal{H}}V)(\tau_h) := \mathbb{E}_{\substack{a_h \sim \pi_e \\ a_{h+1:H} \sim \pi_b}}[r_h + V(f_{h+1}) \mid \tau_h] - \mathbb{E}_{\pi_b}[V(f_h) \mid \tau_h] = \langle \mathbf{b}(\tau_h), \mathcal{B}_h^{\mathcal{S}}V \rangle. \tag{2}$$

The following lemma shows that, ideally, we would want to find $V$ with small $\mathcal{B}^{\mathcal{S}}V$:

**Lemma 1.** *For any $\pi_e$, $\pi_b$, and $V : \mathcal{F} \rightarrow \mathbb{R}$, $J(\pi_e) - \mathbb{E}_{\pi_b}[V(f_1)] = \sum_{h=1}^H \mathbb{E}_{\pi_e}\left[(\mathcal{B}^{\mathcal{S}}V)(s_h)\right].$*

See proof in Appendix C.1. As the lemma shows, any $V$ with small $\mathcal{B}^{\mathcal{S}}V$ (such as $V_{\mathcal{F}}$, since $\mathcal{B}^{\mathcal{S}}V_{\mathcal{F}} \equiv 0$) can be used to estimate $J(\pi_e)$ via $\mathbb{E}_{\pi_b}[V(f_1)]$. But again, $\mathcal{B}^{\mathcal{S}}$ operates on $\mathcal{S}$ which is unobserved, and we turn to its proxy $\mathcal{B}^{\mathcal{H}}$, which are linear measures of $\mathcal{B}^{\mathcal{S}}$ (Eq.2). More concretely, Uehara et al. [2022a] proposed to estimate $\mathbb{E}_{\pi_b}[(\mathcal{B}^{\mathcal{H}}V)^2]$ using an additional helper class $\Xi : \mathcal{H} \rightarrow \mathbb{R}$ to handle the double-sampling issue [Antos et al., 2008, Dai et al., 2018]:

$$\widehat{V} = \underset{V \in \mathcal{V}}{\operatorname{argmin}} \max_{\xi \in \Xi} \sum_{h=1}^H \mathcal{L}_h(V, \xi), \tag{3}$$

where $\mathcal{L}_h(V, \xi) = \mathbb{E}_{\mathcal{D}}[\{\mu(a_h, o_h)(r_h + V(f_{h+1})) - V(f_h)\}\xi(\tau_h) - 0.5\xi^2(\tau_h)]$. Under the following assumptions, the estimator enjoys a finite-sample guarantee:

**Assumption 5** (Realizability). Let $\mathcal{V} \subset (\mathcal{F} \rightarrow \mathbb{R})$ be a finite function class. Assume $V_{\mathcal{F}} \in \mathcal{V}$ for some $V_{\mathcal{F}}$ satisfying Definition 3.

**Assumption 6** (Bellman completeness). Let $\Xi \subset (\mathcal{H} \rightarrow \mathbb{R})$ be a finite function class. Assume $\mathcal{B}^{\mathcal{H}}V \in \Xi$, $\forall V \in \mathcal{V}$.

**Theorem 2.** *Under Assumptions 5 and 6, w.p. $\geq 1 - \delta$,*

$$|J(\pi_e) - \mathbb{E}_{\mathcal{D}}[\widehat{V}(f_1)]| \leq cH \max\{C_{\mathcal{V}} + 1, C_{\Xi}\} \cdot \mathrm{IV}(\mathcal{V})\mathrm{Dr}_{\mathcal{V}}[d^{\pi_e}, d^{\pi_b}]\sqrt{\frac{C_\mu \log \frac{|\mathcal{V}||\Xi|}{\delta}}{n}},$$

*where $c$ is an absolute constant,[3] and $C_{\mathcal{V}} := \max_{V \in \mathcal{V}} \|V\|_\infty$, $C_{\Xi} := \max_{\xi \in \Xi} \|\xi\|_\infty$,*

$$\mathrm{IV}(\mathcal{V}) := \max_h \sup_{V \in \mathcal{V}} \sqrt{\frac{\mathbb{E}_{\pi_b}\left[(\mathcal{B}^{\mathcal{S}}V)(s_h)^2\right]}{\mathbb{E}_{\pi_b}\left[(\mathcal{B}^{\mathcal{H}}V)(\tau_h)^2\right]}}, \quad \mathrm{Dr}_{\mathcal{V}}[d^{\pi_e}, d^{\pi_b}] := \max_h \sup_{V \in \mathcal{V}} \sqrt{\frac{\mathbb{E}_{\pi_e}\left[(\mathcal{B}^{\mathcal{S}}V)(s_h)^2\right]}{\mathbb{E}_{\pi_b}\left[(\mathcal{B}^{\mathcal{S}}V)(s_h)^2\right]}}.$$

---

[2]We let $V(f_{H+1}) \equiv 0$ for all $V$ to be considered, so that we do not need to handle the $H$-th step separately.
[3]The value of $c$ can differ in each occurrence, and we reserve the symbol $c$ for such absolute constants.

See proof in Appendix C.2. Among the objects that appear in the bound, some are mundane and expected (e.g., horizon $H$, the complexities of function classes $\log |\mathcal{V}||\Xi|$, etc.). We now focus on the important ones, which reveals the open questions we shall investigate next:

1. $\mathrm{Dr}_{\mathcal{V}}[d^{\pi_e}, d^{\pi_b}]$ measures the coverage of $\pi_b$ over $\pi_e$ on the latent state space $\mathcal{S}$, as the expression inside square-root can be bounded by $\max_{s_h} d^{\pi_e}(s_h)/d^{\pi_b}(s_h)$.

2. **(Q1)** $C_{\mathcal{V}}$ is the range of the $\mathcal{V}$ class. Since $V_{\mathcal{F}} \in \mathcal{V}$ (Assumption 5), $C_{\mathcal{V}} \geq \|V_{\mathcal{F}}\|_\infty$. However, unlike the standard value functions in MDPs which have obviously bounded range $[0, H]$ (this also applies to $V_{\mathcal{S}}^{\pi_e}$), the range of $V_{\mathcal{F}}$ is unclear. Similarly, since $\Xi$ needs to capture $\mathcal{B}^{\mathcal{H}}V$ for $V \in \mathcal{V}$, the boundedness of $\Xi$ is also affected.

3. **(Q2)** $\mathrm{IV}(\mathcal{V})$ appears because we use $\mathcal{B}^{\mathcal{H}}V$ as a proxy for $\mathcal{B}^{\mathcal{S}}V$, which is only a linear measure of the latter, $(\mathcal{B}^{\mathcal{H}}V)(\tau_h) = \langle \mathbf{b}(\tau_h), \mathcal{B}_h^{\mathcal{S}}V \rangle$. $\mathrm{IV}(\mathcal{V})$ reflects the conversion ratio between them. Uehara et al. [2022a] pointed out that the value is finite if $M_{\mathcal{H},h}$ has full-row rank $(S)$, but a quantitative understanding is missing.

The rest of the paper proposes new coverage assumptions, analyses, and algorithms to answer the above questions, providing deeper understanding on the mathematical structure of OPE in POMDPs.

## 4 Boundedness of FDVFs

We start with **Q1**: when are FDVFs bounded? Recall from Definition 3 that a FDVF at level $h$, $V_{\mathcal{F},h}$, is any function satisfying $M_{\mathcal{F},h} \times V_{\mathcal{F},h} = V_{\mathcal{S},h}^{\pi_e}$, where $M_{\mathcal{F},h}$ is the outcome matrix with $(M_{\mathcal{F},h})_{i,j} = \mathrm{Pr}_{\pi_b}(f_h = j \mid s_h = i)$. Since the equation can have many solutions and we only need to find one of them, it suffices to provide an explicit construction of a $V_{\mathcal{F},h}$ and show it is well bounded. Also note that the equations for different $h$ are independent of each other, so we can construct $V_{\mathcal{F},h}$ for each $h$ separately.

We first describe two natural constructions, both of which yield exponentially large upper bounds.

**Importance Sampling Solution.** Uehara et al. [2022a] provided a cruel bound on $\|V_{\mathcal{F}}\|_\infty$ which scales with $1/\sigma_{\min}(M_{\mathcal{F},h})$, which we show shortly is an exponential-in-length quantity. However, it is not hard to notice that in certain benign cases, $V_{\mathcal{F}}$ does not have to blow-up exponentially and has obviously bounded constructions.

As a warm-up, consider the *on-policy* case of $\pi_b = \pi_e$, where a natural solution is $V_{\mathcal{F}}(f_h) = R^+(f_h) := \sum_{h'=h}^{H} R(o_{h'}, a_{h'})$. Here $R^+(f_h)$ simply adds up the Monte-Carlo rewards in future $f_h$, which is bounded in $[0, H]$. The construction trivially satisfies $\mathbb{E}_{\pi_b}[V_{\mathcal{F}}(f_h) \mid s_h] = V_{\mathcal{S}}^{\pi_e}(s_h)$ when $\pi_b = \pi_e$. In the more general setting of $\pi_b \neq \pi_e$, the hope is that $V_{\mathcal{F}}$ can be still bounded up to some coverage condition that measures how $\pi_e$ deviates from $\pi_b$.

Unfortunately, generalizing the above equation to the off-policy case raises issues: consider the solution $V_{\mathcal{F}}(f_h) = R^+(f_h) \cdot \prod_{h'=h}^{H} \frac{\pi_e(a_{h'}|o_{h'})}{\pi_b(a_{h'}|o_{h'})}$, whose correctness can be verified by importance sampling. Despite its validity, the construction involves cumulative action importance weights, which erases the superiority of the framework over IS as discussed in the introduction.

**Pseudo-inverse Solution.** Another direction is to simply treat Eq.(1) as a linear system. Given that the system is under-determined, we can use pseudo-inverse:[4] $V_{\mathcal{F},h} = M_{\mathcal{F},h}^\top \left( M_{\mathcal{F},h} M_{\mathcal{F},h}^\top \right)^{-1} V_{\mathcal{S},h}^{\pi_e}$. In this case, $\|V_{\mathcal{F}}\|_\infty$ can be bounded using $1/\sigma_{\min}(M_{\mathcal{F},h})$, where $\sigma_{\min}$ denotes the smallest singular value. In fact, closely related quantities have appeared in the recent POMDP literature; for example, in the online setting, Liu et al. [2022a] used $\sigma_{\min}$ of an action-conditioned variant of $M_{\mathcal{F},h}$ as a complexity parameter for online exploration in POMDPs. Unfortunately, these quantities suffer from scaling issues: $1/\sigma_{\min}(M_{\mathcal{F},h})$ is *guaranteed* to be misbehaved if the process is sufficiently stochastic.

**Example 1.** *If* $\mathrm{Pr}_{\pi_b}(f_h|s_h) \leq \frac{C_{stoch}}{(OA)^{H-h+1}}, \forall f_h, s_h$, *then* $\sigma_{\min}(M_{\mathcal{F},h}) \leq C_{stoch}\sqrt{S}/(OA)^{\frac{H-h+1}{2}}$.

---

[4]All covariance-like matrices in the paper, such as $M_{\mathcal{F},h} M_{\mathcal{F},h}^\top$ here, are invertible under Assumption 2.

In this example, $C_{\text{stoch}}$ measures how the distribution of $f_h$ under $\pi_b$ deviates multiplicatively from a uniform distribution over $\mathcal{F}_h$, and an even moderately stochastic $\pi_b$ and emission process $\mathbb{O}$ will lead to small $C_{\text{stoch}}$, which implies an exponentially large $1/\sigma_{\min}(M_{\mathcal{F},h})$.

## 4.1 Minimum Weighted 2-Norm Solution and $L_2$ Outcome Coverage

Pseudo-inverse finds the minimum $L_2$ norm solution. However, given that we are searching for solutions in $\mathbb{R}^{\mathcal{F}_h}$ which has an exponential dimensionality, the standard $L_2$ norm—which treats all coordinates equally—is not a particularly informative metric. Instead, we propose to minimize the *weighted* $L_2$ norm with a particular weighting scheme, which has also been used in HMMs [Mahajan et al., 2023] and enjoys benign properties.

We first define the diagonal weight matrix $Z_h := \text{diag}(\mathbf{1}_{\mathcal{S}}^\top M_{\mathcal{F},h})$, where $\mathbf{1}_{\mathcal{S}} \in \mathbb{R}^{\mathcal{S}} = [1, \cdots, 1]^\top$ is the all-one vector. Then, the solution that minimizes $\|\cdot\|_{Z_h}$ is:

$$V_{\mathcal{F},h} = Z_h^{-1} M_{\mathcal{F},h}^\top \Sigma_{\mathcal{F},h}^{-1} V_{\mathcal{S},h}^{\pi_e}, \quad \text{where } \Sigma_{\mathcal{F},h} := M_{\mathcal{F},h} Z_h^{-1} M_{\mathcal{F},h}^\top. \tag{4}$$

$\Sigma_{\mathcal{F},h} \in \mathbb{R}^{S \times S}$ plays an important role in this construction. Recall that its counterpart in the pseudo-inverse solution, namely $M_{\mathcal{F},h} M_{\mathcal{F},h}^\top$, has scaling issues (Example 1), that even its *largest* eigenvalue can decay exponentially with $H - h + 1$. In contrast, $\Sigma_{\mathcal{F},h}$ is very well-behaved in its magnitude, as shown below. Furthermore, while $M_{\mathcal{F},h}^\top$ on the left is now multiplied by $Z_h^{-1}$ which can be exponentially large, $Z_h^{-1} M_{\mathcal{F},h}^\top$ together is still well-behaved; see proof in Appendix D.2.

**Proposition 3** (Properties of Eq.(4)). *1. $\Sigma_{\mathcal{F},h}$ is doubly-stochastic: that is, each row/column of $\Sigma_{\mathcal{F},h}$ is non-negative and sums up to 1. As a consequence, $\sigma_{\max}(\Sigma_{\mathcal{F},h}) = 1$.*
*2. Rows of $Z_h^{-1} M_{\mathcal{F},h}^\top$, i.e., $\{\mathbf{u}(f_h)^\top / Z(f_h) : f_h \in \mathcal{F}_h\}$, are stochastic vectors, i.e., they are non-negative and the row sum is 1.*

Therefore, it is promising to make the assumption that $\sigma_{\min}(\Sigma_{\mathcal{F},h})$ is bounded away from zero, which immediately leads to the boundedness of $V_{\mathcal{F}}$ given that of $Z_h^{-1} M_{\mathcal{F},h}^\top$ ([0, 1]) and $V_{\mathcal{S}}^{\pi_e}$ ([0, H]). However, we need to rule out the possibility that $\Sigma_{\mathcal{F},h}$ is always near-singular, and find natural examples that admit large $\sigma_{\min}(\Sigma_{\mathcal{F},h})$, as given below.

**Example 2.** *Suppose $f_h$ always reveals $s_h$, in the sense that for any $j \in \mathcal{F}_h$, $\Pr_{\pi_b}(f_h = j \mid s_h = i)$ is only non-zero for a single $i \in [S]$, and zero for all other latent states. Then, $\Sigma_{\mathcal{F},h} = \mathbf{I}$, the identity matrix. Furthermore, $V_{\mathcal{F}}$ from Eq.(4) satisfies $\|V_{\mathcal{F}}\|_\infty \leq H$. See Appendix D.3 for details.*

The example shows an ideal case where $\sigma_{\min}(\Sigma_{\mathcal{F},h}) = \sigma_{\max}(\Sigma_{\mathcal{F},h}) = 1$, when the future fully determines $s_h$. This can happen when the last observation $o_H$ reveals the identity of an earlier latent state $s_h$. Note that in this case, $M_{\mathcal{F},h} M_{\mathcal{F},h}^\top$ can still have poor scaling if the actions and observations between step $h$ and $H$ are sufficiently stochastic, which shows how the weighted 2-norm solution and analysis improve over the pseudo-inverse one. More generally, $\Sigma_{\mathcal{F},h}$ is the confusion matrix of making posterior predictions of $s_h$ from $f_h$ based on a uniform prior over $\mathcal{S}_h$ (see Appendix B.8 for how to incorporate different priors) with $\mathbf{u}(f_h)^\top / Z(f_h)$ being the posterior, and $\sigma_{\min}(\Sigma_{\mathcal{F},h})$ serves as a measure of how the distribution of future $f_h$ helps reveal the latent state $s_h$.

We now break down the boundedness of Eq.(4) into more interpretable assumptions.

**Assumption 7** ($L_2$ outcome coverage). Assume for all $h$, $\|V_{\mathcal{S},h}^{\pi_e}\|_{\Sigma_{\mathcal{F},h}^{-1}}^2 \leq C_{\mathcal{F},V}$.

**Assumption 8** ($\Sigma_{\mathcal{F},h}$ regularity). Assume for any $f_h$: $\|\mathbf{u}(f_h)/Z(f_h)\|_{\Sigma_{\mathcal{F},h}^{-1}}^2 \leq C_{\mathcal{F},U}$.

**Proposition 4** (Boundedness of FDVF). *Under Assumptions 7 and 8, $V_{\mathcal{F}}$ in Eq.4 satisfies $\|V_{\mathcal{F}}\|_\infty \leq \sqrt{C_{\mathcal{F},2}} := \sqrt{C_{\mathcal{F},V} C_{\mathcal{F},U}}$. Furthermore, when only Assumption 7 holds, $\|V_{\mathcal{F},h}\|_{Z_h} \leq \sqrt{C_{\mathcal{F},V}}, \forall h$.*

See proof in Appendix D.4. Assumption 7 requires that the weighted covariance matrix $\Sigma_{\mathcal{F},h}$ covers the direction of $V_{\mathcal{S},h}^{\pi_e}$ well. As a sanity check, it is always bounded in the on-policy case:

**Example 3.** *When $\pi_b = \pi_e$, $\|V_{\mathcal{S},h}^{\pi_e}\|_{\Sigma_{\mathcal{F},h}^{-1}} \leq H\sqrt{S}$.*

Notably, mathematically similar coverage assumptions are also found in the linear MDP literature. For example, with state-action feature $\phi_h$ for time step $h$, a very tight coverage parameter for linear

MDPs is $\|\mathbb{E}_{\pi_e}[\phi_h]\|^2_{\mathbb{E}_{\pi_b}[\phi_h \phi_h^\top]^{-1}}$ [Zanette et al., 2021]. Despite the mathematically similarity, there are important high-level differences between these notions of coverage:

1. As mentioned earlier, MDP coverage is concerned with the dynamics **before** step $h$, whereas our outcome coverage concerns that **after** $h$. Relatedly, MDP coverage depends on the initial distribution (which our outcome coverage does not depend on), and our coverage depends on the reward function through $V_{\mathcal{F}}$ (which MDP coverage does not explicitly depend on). In Section 5, we will discuss our other coverage assumption (belief coverage), which is more similar to the MDP coverage in that they are both concerned with the past.

2. The linear MDP coverage assumption is a refinement of state-density ratio using the knowledge of the function class [Chen and Jiang, 2019, Song et al., 2022]. In comparison, the linear structure of our outcome-coverage assumption comes directly from the internal structure of POMDPs.

### 4.2 Addressing $S$ dependence via $L_1/L_\infty$ Hölder and $L_\infty$ Outcome Coverage

Example 3 shows that even in the on-policy case, $C_{\mathcal{F},V}$ may depend on $S$ which makes the assumption only meaningful for finite and small $\mathcal{S}$. In fact, we showed earlier that $V_{\mathcal{F}} = R^+$ is a natural and obvious $L_\infty$-bounded solution for $\pi_b = \pi_e$, but this is not recovered by the construction in Eq. (4). We also need an additional regularity Assumption 8.

As it turns out, these undesired properties arise because $L_2$ Hölder—which is natural for linear MDP settings mentioned above—fails to leverage the $L_1$ normalization of $\mathbf{u}(f_h)^\top/Z(f_h)$ (Proposition 4, Claim 2) and is loose for POMDPs; see Appendix B.5 for further details. A better choice is $L_1/L_\infty$ Hölder, motivating the $L_\infty$ coverage assumption below, which requires a slightly different construction of $V_{\mathcal{F}}$. These definitions may seem mysterious or even counterintuitive; it will be easier to explain the intuitions when we get to their counterparts for belief coverage in Section 5.2.

**Construction of $V_{\mathcal{F}}$** Define $Z^R(f_h) := Z(f_h)/R^+(f_h)$, and we use $Z^R$ to replace $Z$ in Eq.(4):

$$V_{\mathcal{F},h} = (Z_h^R)^{-1} M_{\mathcal{F},h}^\top (\Sigma_{\mathcal{F},h}^R)^{-1} V_{\mathcal{S},h}^{\pi_e}, \quad \text{where } \Sigma_{\mathcal{F},h}^R := M_{\mathcal{F},h}(Z_h^R)^{-1} M_{\mathcal{F},h}^\top. \tag{5}$$

**Assumption 9** ($L_\infty$ outcome coverage). Assume for all $h$, $\|(\Sigma_{\mathcal{F},h}^R)^{-1} V_{\mathcal{S}}^{\pi_e}\|_\infty \leq C_{\mathcal{F},\infty}$.

**Lemma 5.** *Under Assumption 9, $\|V_{\mathcal{F}}\|_\infty \leq H C_{\mathcal{F},\infty}$. See proof in Appendix E.7.*

In Appendix B.5 we show that the construction shares similar properties to Eq.(4) in the scenario of Example 2. On the other hand, it has better scaling properties w.r.t. $S$ and does not additionally require a regularity assumption like Assumption 8. In the on-policy case, Eq.(5) *exactly* recovers $V_{\mathcal{F}} = R^+$, a property that Eq.(4) does not enjoy; see Appendix E.8 for details.

**Example 4.** *When $\pi_e = \pi_b$, $(\Sigma_{\mathcal{F},h}^R)^{-1} V_{\mathcal{S}}^{\pi_e} = \mathbf{1}$, thus Assumption 9 holds with $C_{\mathcal{F},\infty} = 1$ (c.f. $C_{\mathcal{F},V} \leq H\sqrt{S}$ in Example 3). Furthermore, the construction in Eq.(5) is exactly $V_{\mathcal{F}} = R^+$.*

## 5 Effective History Weights and A New Algorithm

We now turn to **Q2** in Section 3, which asks for a quantitative understanding of the $\mathrm{IV}(\mathcal{V})$ term. Note that $\mathrm{IV}(\mathcal{V})$ and $\mathrm{Dr}_{\mathcal{V}}[d^{\pi_e}, d^{\pi_b}]$, taken together, are to address the conversion between:

$$\text{(We minimize:) } \sqrt{\mathbb{E}_{\pi_b}[(\mathcal{B}^{\mathcal{H}}V)(\tau_h)^2]} \quad \rightarrow \quad \text{(We want to bound:) } |\mathbb{E}_{\pi_e}[(\mathcal{B}^{\mathcal{S}}V)(s_h)]|. \tag{6}$$

The two terms differ both in the policy ($\pi_b \to \pi_e$) and the operator ($\mathcal{B}^{\mathcal{H}} \to \mathcal{B}^{\mathcal{S}}$). While we could directly define a parameter by taking the worst-case (over $V \in \mathcal{V}$) ratio between the two expressions,[5] the real question is to provide more intuitive understanding of when it can be bounded.

Towards this goal, Uehara et al. [2022a] split the above ratio into two terms, $\mathrm{IV}(\mathcal{V})$ and $\mathrm{Dr}_{\mathcal{V}}[d^{\pi_e}, d^{\pi_b}]$, which take care of the $\mathcal{B}^{\mathcal{H}} \to \mathcal{B}^{\mathcal{S}}$ conversion (under $\pi_b$) and $\pi_b \to \pi_e$ conversion (under $\mathcal{B}^{\mathcal{S}}$), respectively. While this leads to an intuitive upper bound of the $\pi_b \to \pi_e$ conversion parameter in terms of latent state coverage, the nature of the $\mathcal{B}^{\mathcal{H}} \to \mathcal{B}^{\mathcal{S}}$ conversion, $\mathrm{IV}(\mathcal{V})$, remains mysterious.

---

[5]Recent offline RL theory indeed employed such definitions [Xie et al., 2021, Song et al., 2022], but that is after we developed mature understanding of their properties, such as being upper bounded by density ratios.

In this section, we take a different approach by *directly* providing an intuitive upper bound on both conversions altogether, under a novel *belief coverage* assumption. In Appendix E.4 we will also revisit the split into $\mathrm{IV}(\mathcal{V})$ and $\mathrm{Dr}_{\mathcal{V}}[d^{\pi_e}, d^{\pi_b}]$: in the absence of strong structures from $\mathcal{V}$, the boundedness of $\mathrm{IV}(\mathcal{V})$ turns out to require an even stronger version of belief coverage, rendering the split unnecessary. Furthermore, our approach also leads to a novel algorithm for estimating $J(\pi_e)$ that replaces Bellman-completeness (Assumption 6) with a weight-realizability assumption, similar to MIS estimators for MDPs [Liu et al., 2018, Uehara et al., 2020].

## 5.1 Effective History Weights

The key idea in this section is the notion of *effective history weights*, that perform the $\pi_b \to \pi_e$ and $\mathcal{B}^{\mathcal{H}} \to \mathcal{B}^{\mathcal{S}}$ conversions jointly.

**Definition 10** (Effective history weights). An effective history weight function $w^{\star} : \mathcal{H} \to \mathbb{R}$ is any function that satisfies: $\forall V \in \mathcal{V}$, $h \in [H]$,

$$\mathbb{E}_{\pi_b}[w^{\star}(\tau_h)(\mathcal{B}^{\mathcal{H}}V)(\tau_h)] = \mathbb{E}_{\pi_e}\left[(\mathcal{B}^{\mathcal{S}}V)(s_h)\right]. \tag{7}$$

We first see that well-bounded $w^{\star}$ immediately leads to a good conversion ratio for Eq.(6):

$$|\mathbb{E}_{\pi_e}\left[(\mathcal{B}^{\mathcal{S}}V)(s_h)\right]| = |\mathbb{E}_{\pi_b}[w^{\star}(\tau_h)(\mathcal{B}^{\mathcal{H}}V)(\tau_h)]| \le \sqrt{\mathbb{E}_{\pi_b}[w^{\star}(\tau_h)^2]\mathbb{E}_{\pi_b}[(\mathcal{B}^{\mathcal{H}}V)(\tau)^2]},$$

where the inequality follows from Cauchy-Schwartz for r.v.'s. Hence, all we need is $\|w^{\star}\|_{2,d_h^{\pi_b}} := \sqrt{\mathbb{E}_{\pi_b}[w^{\star}(\tau_h)^2]}$, the $\pi_b$-weighted 2-norm of $w^{\star}$, to be bounded, and we focus on this quantity next. Similar to Section 4, there may be multiple $w^{\star}$ that satisfies the definition, and we only need to show the boundedness of *any* solution. Also similarly, the most obvious solution is $w^{\star}(\tau_h) = \frac{\Pr_{\pi_e}(\tau_h)}{\Pr_{\pi_b}(\tau_h)} = \prod_{h'=1}^{h-1} \frac{\pi_e(a_{h'}|o_{h'})}{\pi_b(a_{h'}|o_{h'})}$, noting that $\mathbb{E}_{\pi_e}\left[(\mathcal{B}^{\mathcal{S}}V)(s_h)\right] = \mathbb{E}_{\pi_e}\left[(\mathcal{B}^{\mathcal{H}}V)(\tau_h)\right]$ and importance weighting on $\tau_h$ changes $\mathbb{E}_{\pi_b}$ to $\mathbb{E}_{\pi_e}$. However, the use of cumulative importance weights is undesirable given its exponential nature, causing the history-version of "curse of horizon" Liu et al. [2018].

**Construction by Belief Matching.** We now show a better construction that is bounded under a natural *belief coverage* assumption. Note that Def 10 can be written as:

$$\mathbb{E}_{\pi_b}[w^{\star}(\tau_h)\langle \mathbf{b}(\tau_h), \mathcal{B}_h^{\mathcal{S}}V\rangle] = \mathbb{E}_{\pi_e}\left[\langle \mathbf{b}(\tau_h), \mathcal{B}_h^{\mathcal{S}}V\rangle\right],$$

where $\mathcal{B}_h^{\mathcal{S}}V \in \mathbb{R}^S$ is the Bellman residual vector for $V$ on $\mathcal{S}_h$. As a sufficient condition (which is also necessary when $\mathcal{V}$ lacks strong structures, i.e., $\{\mathcal{B}_h^{\mathcal{S}}V : V \in \mathcal{V}\}$ spans the entire $\mathbb{R}^S$), we can find $w^{\star}$ that satisfies: $\mathbb{E}_{\pi_b}[w^{\star}(\tau_h)\mathbf{b}(\tau_h)] = \mathbb{E}_{\pi_e}\left[\mathbf{b}(\tau_h)\right] =: \mathbf{b}_h^{\pi_e}$. This is related to the mean matching problem in the distribution shift literature [Gretton et al., 2009, Yu and Szepesvári, 2012], and a standard solution is [Bruns-Smith et al., 2023]:

$$w^{\star}(\tau_h) = \mathbf{b}(\tau_h)^{\top}\Sigma_{\mathcal{H},h}^{-1}\mathbf{b}_h^{\pi_e}, \tag{8}$$

where $\Sigma_{\mathcal{H},h} := \sum_{\tau_h} d^{\pi_b}(\tau_h)\mathbf{b}(\tau_h)\mathbf{b}(\tau_h)^{\top}$, and $\mathbf{b}_h^{\pi_e}$ coincides with $[d^{\pi_e}(s_h)]_{s_h}$.

The weighted 2-norm of this solution is immediately bounded under the following assumption.

**Assumption 11** ($L_2$ belief coverage). Assume $\forall h$, $\|\mathbf{b}_h^{\pi_e}\|_{\Sigma_{\mathcal{H},h}^{-1}}^2 \le C_{\mathcal{H},2}$.

**Lemma 6.** *Under Assumption 11, $w^{\star}$ in Eq. (8) satisfies $\forall h$, $\|w^{\star}\|_{2,d_h^{\pi_b}}^2 \le C_{\mathcal{H},2}$. See Appendix E.1.*

Assumption 11 requires that the covariance matrix of belief states under $\pi_b$ covers $\mathbf{b}^{\pi_e}$, the average belief state under $\pi_e$. As before, we also check the on-policy case (see Appendix E.2):

**Example 5.** *In the on-policy case ($\pi_b = \pi_e$), $C_{\mathcal{H},2} \le 1$.*

**Algorithm Guarantee** Now we have all the pieces to present a fully polynomial version of Theorem 2 under the proposed coverage assumptions. One subtlety is that the guarantee depends on $C_{\mathcal{V}} := \max_{V \in \mathcal{V}} \|V\|_{\infty}$, which is closely related to $\|V_{\mathcal{F}}\|_{\infty}$ since we require $V_{\mathcal{F}} \in \mathcal{V}$, but they are not equal since $\mathcal{V}$ can include other functions with higher range. To highlight the dependence of $\|V_{\mathcal{F}}\|_{\infty}$ on the proposed coverage assumptions, we follow Xie and Jiang [2020] to assume that the range of the function classes is not much larger than that of the function it needs to capture. A similar assumption applies for bounding $C_{\Xi}$. The proof of the theorem is deferred to Appendix E.10.

**Theorem 7.** *Consider the same setting as Theorem 2, and let Assumptions 11 and 9 hold. For some absolute constant $c$, further assume $C_{\mathcal{V}} \leq c\|V_{\mathcal{F}}\|_{\infty}$ for $V_{\mathcal{F}}$ in Eq.(5) and $C_{\Xi} \leq c(\|V_{\mathcal{F}}\|_{\infty} + 1)$. W.p. $\geq 1 - \delta$, $|J(\pi_e) - \mathbb{E}_{\mathcal{D}}[\widehat{V}(f_1)]| \leq cH^2(C_{\mathcal{F},\infty} + 1)\sqrt{\frac{C_{\mathcal{H},2}C_{\mu}\log(|\mathcal{V}||\Xi|/\delta)}{n}}$.*

In addition to $H$ and complexities of $|\mathcal{V}|$ and $|\Xi|$, the bound *only* depends on the intuitive coverage parameters: $C_{\mu}$ (action coverage), $C_{\mathcal{H},2}$ ($L_2$ belief coverage), and $C_{\mathcal{F},\infty}$ ($L_{\infty}$ outcome coverage).

## 5.2 $L_{\infty}$ Belief Coverage

Assumption 11 enables bounded second moment of $w^{\star}$ (Lemma 6) but does not control $\|w^{\star}\|_{\infty}$, which we show will be useful for the new algorithm in Section 5.3. Here we present the $L_{\infty}$ version of belief coverage that controls $\|w^{\star}\|_{\infty}$, which also helps understand $L_{\infty}$ outcome coverage given the symmetry between history and future. As alluded to in Section 4.2 and Appendix B.5, $L_2$ Hölder is inappropriate for controlling the infinity-norm since it does not leverage the $L_1$ normalization of vectors in POMDPs. Instead, we propose the following decomposition based on $L_1/L_{\infty}$ Hölder: $|w^{\star}(\tau_h)| \leq \|\mathbf{b}(\tau_h)\|_1 \|\Sigma_{\mathcal{H},h}^{-1}\mathbf{b}_h^{\pi_e}\|_{\infty} = \|\Sigma_{\mathcal{H},h}^{-1}\mathbf{b}_h^{\pi_e}\|_{\infty}$. This way, we can immediately bound $\|w^{\star}\|_{\infty}$ with the following assumption:

**Assumption 12** ($L_{\infty}$ belief coverage). *Assume $\forall h$, $\|\Sigma_{\mathcal{H},h}^{-1}\mathbf{b}_h^{\pi_e}\|_{\infty} \leq C_{\mathcal{H},\infty}$. Then $\|w^{\star}\|_{\infty} \leq C_{\mathcal{H},\infty}$.*

$\Sigma_{\mathcal{H},h}^{-1}\mathbf{b}_h^{\pi_e}$ is the inverse of *second* moment (covariance) multiplying the *first* moment (expectation), raising the concern that the quantity may be poorly scaled: for example, given bounded (but otherwise arbitrary) random vector $X$, $\mathbb{E}[XX^{\top}]^{-1}\mathbb{E}[X]$ can go to infinity if we rescale $X$ by a small constant. First note that such a pathology cannot happen here because the random vectors $(\mathbf{b}(\tau_h))$ are $L_1$-normalized and cannot be arbitrarily rescaled. Below we use a few examples to show that $\Sigma_{\mathcal{H},h}^{-1}\mathbf{b}_h^{\pi_e}$ is a very well-behaved quantity, and naturally generalize familiar concepts such as *concentrability coefficient* from MDPs [Munos, 2007, Chen and Jiang, 2019, Uehara et al., 2022a].

We start by checking the on-policy case. Perhaps surprisingly, $\Sigma_{\mathcal{H},h}^{-1}\mathbf{b}_h^{\pi_e}$ has an *exact* solution:

**Example 6.** *When $\pi_e = \pi_b$, $\Sigma_{\mathcal{H},h}^{-1}\mathbf{b}_h^{\pi_e} = \mathbf{1}$, the all-one vector; see Appendix E.5 for the calculation. Consequently, Assumption 12 is satisfied with $C_{\mathcal{H},\infty} = 1$.*

The next scenario considers when $\mathbf{b}(\tau_h)$ is always one-hot, i.e., histories reveal the latent state (this is analogous to Example 2. $L_{\infty}$ coverage reduces to the familiar *concentrability coefficient*, the infinity-norm of density ratio as a standard coverage parameter:

**Example 7.** *When $\mathbf{b}(\tau_h)$ is always one-hot, $\|\Sigma_{\mathcal{H},h}^{-1}\mathbf{b}_h^{\pi_e}\|_{\infty} = \max_{s_h} d^{\pi_e}(s_h)/d^{\pi_b}(s_h)$.*

We can also calculate $\|\mathbf{b}_h^{\pi_e}\|_{\Sigma_{\mathcal{H},h}^{-1}}^2$ from Assumption 11, which equals $\mathbb{E}_{\pi_b}[(d^{\pi_e}(s_h)/d^{\pi_b}(s_h))^2]$ in this "1-hot belief" scenario. Xie and Jiang [2020] show that this is tighter than $\max_{s_h} d^{\pi_e}(s_h)/d^{\pi_b}(s_h)$, and this relation extends elegantly to general belief vectors in our setting:

**Lemma 8.** $\|\mathbf{b}_h^{\pi_e}\|_{\Sigma_{\mathcal{H},h}^{-1}}^2 \leq \|\Sigma_{\mathcal{H},h}^{-1}\mathbf{b}_h^{\pi_e}\|_{\infty}$. *See proof in Appendix E.6.*

## 5.3 New Algorithm

The discovery of effective history weights and its boundedness also lead to a new algorithm analogous to MIS methods for MDPs [Uehara et al., 2020]. The idea is that since Lemma 1 tells us to minimize $\mathbb{E}_{\pi_e}[(\mathcal{B}^{\mathcal{S}}V)(s_h)]$, which equals $\mathbb{E}_{\pi_b}[w^{\star}(\tau_h)(\mathcal{B}^{\mathcal{H}}V)(\tau_h)]$ from Def 10, we can then use another function class $\mathcal{W} \subset (\mathcal{H} \to \mathbb{R})$ to model $w^{\star}$, and approximately solve the following:

$$\underset{V \in \mathcal{V}}{\operatorname{argmin}} \max_{w \in \mathcal{W}} \sum_{h=1}^{H} |\mathbb{E}_{\pi_b}[w(\tau_h)(\mathcal{B}^{\mathcal{H}}V)(\tau_h)]|, \tag{9}$$

which minimizes an upper bound of $\mathbb{E}_{\pi_b}[w^{\star}(\tau_h)(\mathcal{B}^{\mathcal{H}}V)(\tau_h)]$ as long as $w^{\star} \in \mathcal{W} \subset (\mathcal{H} \to \mathbb{R})$. Since there is no square inside the expectation, there is no double-sampling issue and we thus do not need the $\Xi$ class and its Bellman-completeness assumption. The $h$-th term of the loss can be estimated straightforwardly as

$$|\mathbb{E}_{\mathcal{D}}[w(\tau_h)(\mu(o_h, a_h)(r_h + V(f_{h+1})) - V(f_h))]|. \tag{10}$$

We now provide the sample-complexity analysis of the algorithm, using a more general analysis that allows for approximation errors in $\mathcal{V}$ and $\mathcal{W}$.

**Assumption 13** (Approximate realizablity). Assume

$$\min_{V \in \mathcal{V}} \max_{w \in \mathcal{W}} \left| \sum_{h=1}^{H} \mathbb{E}_{\pi_b} \left[ w_h(\tau_h) \cdot (\mathcal{B}^{\mathcal{H}} V)(\tau_h) \right] \right| \leq \epsilon_{\mathcal{V}},$$

$$\inf_{w \in \mathrm{sp}(\mathcal{W})} \max_{V \in \mathcal{V}} \left| \sum_{h=1}^{H} \mathbb{E}_{\pi_b} \left[ (w_h^\star(\tau_h) - w_h(\tau_h)) \cdot (\mathcal{B}^{\mathcal{H}} V)(\tau_h) \right] \right| \leq \epsilon_{\mathcal{W}}.$$

Instead of measuring how $\mathcal{W}$ and $\mathcal{V}$ capture our specific constructions of $V_{\mathcal{F}}$ and $w^\star$, the above approximation errors automatically allow all possible solutions by measuring the violation of the equations that define $V_{\mathcal{F}}$ and $w^\star$.

We present the sample complexity bound of our algorithm as follows. Similar to Theorem 7, we assume that $C_{\mathcal{V}} := \max_{V \in \mathcal{V}} \|V\|_\infty$ and $C_{\mathcal{W}} := \max_h \sup_{w \in \mathcal{W}} \|w\|_\infty$ are not much larger than the corresponding norms of $V_{\mathcal{F}}$ and $w^\star$, respectively. See the proof in Appendix E.11.

**Theorem 9.** *Let $\widehat{V}$ be the result of approximating Eq.9 with empirical estimation in Eq.(10). Assume that $C_{\mathcal{V}} \leq c\|V_{\mathcal{F}}\|_\infty$ for $V_{\mathcal{F}}$ in Eq.(5), and $C_{\mathcal{W}} \leq c\|w^\star\|_\infty$ for $w^\star$ in Eq.(8). Under Assumptions 12, 9, and 13, w.p. $\geq 1 - \delta$, $\left| J(\pi_e) - \mathbb{E}_{\mathcal{D}}[\widehat{V}(f_1)] \right| \leq \epsilon_{\mathcal{V}} + \epsilon_{\mathcal{W}} + cH^2 C_{\mathcal{H},\infty}(C_{\mathcal{F},\infty} + 1)\sqrt{\frac{C_\mu \log \frac{|\mathcal{V}||\mathcal{W}|}{\delta}}{n}}$.*

As a remark, there is also a way to leverage the tighter $L_2$ belief coverage, despite that it does not guarantee bounded $\|w^\star\|_\infty$ and only $\|w^\star\|_{2,d_h^{\pi_b}}^2$. In particular, if all functions in $\mathcal{W}$ have bounded $\|\cdot\|_{2,d_h^{\pi_b}}^2$, the estimator in Eq.(10) will have bounded 2nd moment on the data distribution. In this case, using Median-of-Means estimators [Lerasle, 2019, Chen, 2020] instead of plain averages for Eq.(10) will only pay for the 2nd moment and not the range.

# 6 Conclusion and Future Work

The main text considers memoryless policies. Similar to Uehara et al. [2022a], we can extend to policies that depend on recent observations and actions (or *memory*). In fact, we provide a more general result in Appendix B.6 that handles recurrent policies that are *finite state machines*, which allows the policy to depend on long histories. However, the coverage coefficient will be diluted quickly when the memory contains rich information, which we call the *curse of memory*. We suspect that structural policies are needed to avoid the curse of memory and leave this to future work.

## Acknowledgements

Nan Jiang acknowledges funding support from NSF IIS-2112471, NSF CAREER IIS-2141781, Google Scholar Award, and Sloan Fellowship.

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

# A Related Literature

**OPE in POMDPs.** There is a line of research on OPE in confounded POMDPs [Zhang and Bareinboim, 2016, Namkoong et al., 2020, Nair and Jiang, 2021, Guo et al., 2022, Lu et al., 2022, Shi et al., 2022, Hong et al., 2023]. However, all these methods require the behavior policy to solely depend on the latent state. This assumption is inapplicable for our unconfounded POMDP setting, where the behavior policy depends on the observations. Hu and Wager [2023] studied the same unconfounded setting as ours. Nonetheless, their method uses multi-step importance sampling, leading to an undesirable exponential dependence on the horizon. The closest related work to our paper is [Uehara et al., 2022a], which we analyze in detail in Section 3.

**Online Learning in POMDPs.** There are many prior works studying online learning algorithms for sub-classes of POMDPs, including decodable POMDPs [Krishnamurthy et al., 2016, Jiang et al., 2017, Du et al., 2019, Efroni et al., 2022], Latent MDPs [Kwon et al., 2021] and linear quadratic Gaussian setting (LQG) [Lale et al., 2021, Simchowitz et al., 2020]. Recently, online learning algorithms with polynomial sample results have been proposed for both tabular POMDPs [Guo et al., 2016, Azizzadenesheli et al., 2016, Jin et al., 2020, Liu et al., 2022a] and POMDPs with general function approximation [Zhan et al., 2022, Liu et al., 2022b, Chen et al., 2022, Huang et al., 2023]. All of these approaches are model-based and require certain model assumptions. Uehara et al. [2022b] proposed a model-free online learning algorithm based on future-dependent value functions (FDVFs). However, their algorithm requires the boundness of FDVFs for all policies in the policy class, as well as a low-rank property of the Bellman loss, which limits the generality of the results.

**OPE in MDPs.** There has been a long history of studying OPE in MDPs, including importance sampling (IS) approaches [Precup et al., 2000, Li et al., 2011] and their doubly robust variants [Dudík et al., 2011, Jiang and Li, 2016, Thomas and Brunskill, 2016, Farajtabar et al., 2018], marginalized importance sampling (MIS) methods [Liu et al., 2018, Xie et al., 2019, Kallus and Uehara, 2020] and model-based estimators [Eysenbach et al., 2020, Yin et al., 2021, Voloshin et al., 2021]. As mentioned in Section 1, directly applying IS-based approaches for POMDPs will result in exponential variance. Meanwhile, MIS-based approaches do not apply to POMDPs since they require the environment to satisfy the Markov assumption.

# B Discussions and Extensions

## B.1 Connection between Future-dependent Value Functions and Learnable Future-dependent Value Functions

Uehara et al. [2022a] proposed a learnable version of future dependent value functions and focused on the learning of learnable FDVFs instead of the original FDVFs defined in Definition 3. In this subsection, we first introduce the definition of learnable FDVFs and then show that it is equivalent to FDVFs under a full-rank assumption.

**Definition 14** (Learnable future-dependent value functions). A Learnable future-dependent value function $V_{\mathcal{F}}^L : \mathcal{F} \to \mathbb{R}$ is any function that satisfies the following: $\forall \tau_h$,

$$(\mathcal{B}^{\mathcal{H}} V_{\mathcal{F}}^L)(\tau_h) = 0.$$

For any FDVF $V_{\mathcal{F}}$, since $\mathcal{B}^{\mathcal{S}} V_{\mathcal{F}} \equiv 0$, we have $\forall \tau_h$,

$$(\mathcal{B}^{\mathcal{H}} V_{\mathcal{F}})(\tau_h) = \mathbb{E}_{\pi_b}[\mu(o_h, a_h)\{r_h + V_{\mathcal{F}}(f_{h+1})\} - V_{\mathcal{F}}(f_h) \mid \tau_h]$$
$$= \langle \mathbf{b}(\tau_h), \mathcal{B}_h^{\mathcal{S}} V_{\mathcal{F}} \rangle = 0.$$

Therefore, all FDVFs are also learnbale FDVFs. On the other hand, for any learnbale FDVF $V_{\mathcal{F}}^L$, let $\mathcal{B}_h^{\mathcal{H}} V_{\mathcal{F}}^L \in \mathbb{R}^{|\mathcal{H}_h|}$ be the Bellman residual vector for $V_{\mathcal{F}}^L$ on $\mathcal{H}_h$ and $\mathcal{B}_h^{\mathcal{S}} V_{\mathcal{F}}^L \in \mathbb{R}^S$ be the Bellman residual vector on $\mathcal{S}_h$. With the help of $M_{\mathcal{H},h}$, we have the following relation between $\mathcal{B}_h^{\mathcal{H}} V_{\mathcal{F}}^L$ and $\mathcal{B}_h^{\mathcal{S}} V_{\mathcal{F}}^L$:

$$\mathcal{B}_h^{\mathcal{H}} V_{\mathcal{F}}^L = (M_{\mathcal{H},h})^\top \mathcal{B}_h^{\mathcal{S}} V_{\mathcal{F}}^L.$$

When $\mathrm{rank}(M_{\mathcal{H},h}) = S$, $\mathcal{B}_h^{\mathcal{H}} V_{\mathcal{F}}^L = \mathbf{0}$ if and only if $\mathcal{B}_h^{\mathcal{S}} V_{\mathcal{F}}^L = \mathbf{0}$. Therefore, $V_{\mathcal{F}}^L$ satisfies that $\forall s_h, (\mathcal{B}^{\mathcal{S}} V_{\mathcal{F}}^L)(s_h) = 0$, implying that $V_{\mathcal{F}}^L$ is also a FDVF. Under the regular full rank assumption, the two definitions are actually equivalent.

## B.2 Comparison between the Finite-horizon and the Discounted Formulations

The original results of Uehara et al. [2022a] were given in the infinite-horizon discounted setting. To describe their data collection assumption in simple terms, consider an infinite trajectory that enters the stationary configuration of $\pi_b$, and we pick a random time step and call it $t = 0$ (which corresponds to a time step $h$ in our setting). Then, history (analogous to our $\tau_h$) is $(o_{(-M_F):(-1)}, a_{(-M_F):(-1)})$, and future is $(o_{0:M_H}, a_{0:M_F-1})$. $M_F$ and $M_H$ are hyperparameters that determines the lengths, and should not be confused with our $M_{\mathcal{F},h}$ and $M_{\mathcal{H},h}$ matrices. Data points collected in this way form the "transition" dataset. Separately, there is an "initial" dataset of $(o_{0:M_F-1}, a_{0:M_F})$ tuples to represent the analogue of initial state distribution in MDPs.

Besides the fact that finite-horizon formulation better fits real-world applications with episodic nature (e.g., a session in conversational system), there are mathematical reasons why the finite-horizon formulation is more natural:

1. Unlike the infinite-horizon setting, we do not need separate "trainsition" and "initial" datasets. The natural $H$-step episode dataset plays both roles. We also do not need the stationarity assumption.

2. In the infinite-horizon, $M_F$ and $M_H$ are hyperparameters which do not have an obvious upper bound. The larger they are, it is easier to satisfy certain identification assumptions (such as full-rankness of the counterparts of our $M_{\mathcal{H},h}$ and $M_{\mathcal{F},h}$). On the other hand, the "curses of future and horizon" we identified in this work corresponds to exponential dependence on $M_F$ and $M_H$, so there is a potential trade-off in the choice of $M_F$ and $M_H$. In contrast, the future and the history have maximum lengths ($H - h$ and $h - 1$ at time step $h$) in our setting. As we establish coverage assumptions that avoid potential exponential dependencies on these lengths, the trade-off in the choice of length is eliminated, so it is safe for us to always choose the maximum length, as we do in the paper.

## B.3 Refined Coverage

As alluded to at the beginning of Section 5, we can tighten the definition of $L_2$ belief coverage by directly using the ratio:

$$\sup_{V \in \mathcal{V}} \frac{|\mathbb{E}_{\pi_e}[(\mathcal{B}^{\mathcal{S}}V)(s_h)]|}{\sqrt{\mathbb{E}_{\pi_b}[(\mathcal{B}^{\mathcal{H}}V)(\tau_h)^2]}},$$

which automatically leverages the structure of $\mathcal{V}$. For example, if $\{\mathcal{B}_h^{\mathcal{S}}V : V \in \mathcal{V}\}$ only occupies a low-dimensional subspace of $\mathbb{R}^{\mathcal{S}_h}$ (that is, there exists $\phi : \mathcal{S}_h \to \mathbb{R}^d$ such that $(\mathcal{B}^{\mathcal{S}}V)(s_h) = \phi(s_h)^\top \theta_V$), then belief matching above Eq.(8) can be done in $\mathbb{R}^d$ instead of $\mathbb{R}^{\mathcal{S}_h}$. Other coverage definitions may be refined in similar manners.

## B.4 Interpretation of Assumption 8

To interpret Assumption 8, let $x = \mathbf{u}(f_h)/Z(f_h) \in \Delta(\mathcal{S})$, and assign a distribution over $x$ by sampling $f_h \sim \text{diag}(Z_h/S)$, then Assumption 8 becomes $x^\top \mathbb{E}_{Z_h/S}[xx^\top]^{-1}x \leq C_{\mathcal{F},U}S$ for any $x$ with non-zero probability. This is a common regularity assumption [Duan et al., 2020, Perdomo et al., 2023], requiring that no $x$ points to a direction in $\mathbb{R}^S$ alone without being joined by others from the distribution. While $1/\sigma_{\min}(\Sigma_{\mathcal{F},h})$ can bound this quantity, the other way around is not true, implying that Assumption 8 is weaker.

**Example 8** (Bounded $C_{\mathcal{F},U}$ without bounded $1/\sigma_{\min}(\Sigma_{\mathcal{F},h})$)**.** *Consider a distribution over $x$ where with 0.5 probability, $x = [1/2+\epsilon, 1/2-\epsilon]^\top$, and with the other 0.5 probability, $x = [1/2-\epsilon, 1/2+\epsilon]^\top$. Then, as $\epsilon \to 0$, $1/\sigma_{\min}(\Sigma_{\mathcal{F},h}) \to \infty$, but $x^\top \mathbb{E}[xx^\top]^{-1}x \leq 2$.*

## B.5 Intuition for $L_\infty$ Outcome Coverage

Here we provide more details about the $L_\infty$ outcome coverage in Section 4.2, which are omitted in the main text due to space limit.

**Looseness of $L_2$ outcome coverage**  We start by examining the looseness of $L_2$ outcome coverage (Assumption 7), which will motivate the $L_\infty$ version of outcome coverage. To develop intuitions, we

first examine the boundedness of Eq.(4) (the construction that leads to $L_2$ outcome coverage) in the setting of Example 2. In this example, $\Sigma_{\mathcal{F},h} = \mathbf{I}$, and $\|V_{\mathcal{F}}\|_\infty \leq H$. According to the $L_2$ Hölder decomposition in Proposition 4, however,

$$\|V_{\mathcal{F}}\|_\infty \leq \max_{f_h} \|\mathbf{u}(f_h)/Z(f_h)\|_2 \|V_{\mathcal{S}}^{\pi_e}\|_2 \leq H\sqrt{S},$$

where $\|\cdot\|_{\Sigma_{\mathcal{F},h}^{-1}}$ is replaced by $\|\cdot\|_2$ since $\Sigma_{\mathcal{F},h} = I$. In contrast, the tight analysis is

$$\|V_{\mathcal{F}}\|_\infty \leq \max_{f_h} \|\mathbf{u}(f_h)/Z(f_h)\|_1 \|V_{\mathcal{S}}^{\pi_e}\|_\infty \leq H.$$

The comparison clearly highlights that the looseness in $S$ comes from the fact that $L_2$ Hölder does not fully leverage the fact that $\|\mathbf{u}(f_h)/Z(f_h)\|_1 = 1$ and loosely relaxing it to $\|\mathbf{u}(f_h)/Z(f_h)\|_2 \leq 1$. In contrast, our $L_\infty$ coverage assumption allows for a more natural $L_1/L_\infty$ Hölder to leverage the $L_1$ normalization of $\mathbf{u}(f_h)/Z(f_h)$.

$L_\infty$ **Outcome Coverage**   Similar to $L_\infty$ belief coverage, we could define $L_\infty$ outcome coverage simply as $\|(\Sigma_{\mathcal{F},h})^{-1}V_{\mathcal{S}}^{\pi_e}\|_\infty$. The small caveat and inelegance is that it does not recover $V_{\mathcal{F}} = R^+$ when $\pi_b = \pi_e$. To address this, we leverage the lesson from belief coverage (Section 5.2): to obtain $C_{\mathcal{H},\infty} = 1$ when $\pi_b = \pi_e$, a key property is that $\Sigma_{\mathcal{H},h}$ (data covariance matrix) and $\mathbf{b}_h^{\pi_b}$ (the vector to be covered) are the covariance matrix and the mean vector w.r.t. the same distribution of belief vectors. In contrast, for the outcome coverage case, $V_{\mathcal{S}}^{\pi_e}$ depends on the reward function, but such information is missing in $\Sigma_{\mathcal{F},h}$, which prevents a perfect cancellation in the on-policy case.

Therefore, we can adjust the definition of $\Sigma_{\mathcal{F},h}$ to mimic the situation of belief coverage. The first step is to find the counterpart of belief state which is $L_1$-normalized. This obviously corresponds to $Z_h^{-1}M_{\mathcal{F},h}^\top$, whose rows sum up to 1. Let $\bar{\mathbf{u}}(f_h) := \mathbf{u}(f_h)/Z(f_h) \in \Delta(\mathcal{S}_h)$, then $\Sigma_{\mathcal{F},h} = S \cdot \mathbb{E}_{Z_h/S}[\bar{\mathbf{u}}\bar{\mathbf{u}}^\top]$. Then, by incorporating reward information into $\Sigma_{\mathcal{F},h}$, we arrive at the solution in Eq.(5).

Finally, we examine the setting of Example 2 and show that $V_{\mathcal{F}}$ in Eq.(5) is similarly well-behaved as Eq.(4) in this scenario.

**Example 9.** *In the same setting as Example 2, i.e., $f_h$ always reveals $s_h$, we have $\Sigma_{\mathcal{F},h}^R = diag(V_{\mathcal{S}_h}^{\pi_b})$, and $\|(\Sigma_{\mathcal{F},h}^R)^{-1}V_{\mathcal{S}}^{\pi_e}\|_\infty = \max_{s_h} V_{\mathcal{S}}^{\pi_e}(s_h)/V_{\mathcal{S}}^{\pi_b}(s_h)$.*

This example also closely resembles Example 7 for belief coverage. While $(\Sigma_{\mathcal{F},h}^R)^{-1}$ may behave poorly if $V_{\mathcal{S}}^{\pi_b}(s_h)$ is small, an easy fix is to simply add a constant to the reward function and shift its range to e.g., $[1, 2]$, which ensures a small $C_{\mathcal{F},\infty}$.

### B.6   Extension to History-dependent Policies

The main text assumes memoryless $\pi_b$ and $\pi_e$. Similar to Uehara et al. [2022a], we can extend the approach to handle history-dependent policies (the changes to the proof are sketched below). Instead of rewriting the proof with these changes, we will provide a "black-box" reduction that handles the extension more elegantly and allow for more general results than Uehara et al. [2022a].

**Memory-based Policies and Changes in the Proofs**   Define *memory* $m_h$ as a function of $\tau_h$ (i.e., $m_h = m_h(\tau_h)$), and a memory-based policy can depend on $(m_h, o_h)$. Uehara et al. [2022a] allows for $m_h = (o_{h-M:h-1}, a_{h-M:h-1})$ for some fixed window $M$ in their analyses. If we let $m_h = (o_{1:h-1}, a_{1:h-1})$, we recover fully general history-dependent policies.

If we direct modify the proofs to accommodate this generalization (as done in Uehara et al. [2022a]), the required changes are:

- $V_{\mathcal{S}}^{\pi_e}$ and $V_{\mathcal{F}}$ need to additionally depend on $m_h$ to be well defined. $V \in \mathcal{V}$ also generally depend on $m_h$ since we need realizability $V_{\mathcal{F}} \in \mathcal{V}$.

- $M_{\mathcal{F},h}$ and $M_{\mathcal{H},h}$ have rows indexed by $(s_h, m_h)$ instead of just $s_h$. That is, we replace belief state with posterior over $(s_h, m_h)$. Similarly, entries in $M_{\mathcal{F},h}$ are now future probabilities conditioned on $(s_h, m_h)$

However, if we consider latent state coverage, which is weaker than belief coverage that is really needed (Appendix E.3), we now require bounded $\frac{d^{\pi_e}(s_h, m_h)}{d^{\pi_b}(s_h, m_h)}$. This is generally exponential in the length of $m_h$ (e.g., if $M = h - 1$, the ratio is exactly the cumulative importance weight), preventing us from handling $\pi_b$ and $\pi_e$ with long-range history dependencies.

**Reduction to Memoryless Case**    Instead of changing the proofs, we now describe an alternative approach that (1) produces results similar to Uehara et al. [2022a], and (2) handles more general recurrent policies that are *finite-state-machines* (FSMs), which subsume fully history-dependent policies (or policies that depend on a fixed-length window) as special cases.

Concretely, a policy with memory $m_h$ is said to be an FSM, if $m_{h+1}$ can be computed solely based on $m_h, o_h, a_h$, without using other information in $\tau_h$. $m_h = (o_{h-M:h-1}, a_{h-M:h-1})$ satisfies this definition, as computing $m_{h+1}$ is simply dropping the oldest observation-action pair from $m_h$ and appending the newest one.[6] Another example is belief update, where $\mathbf{b}(\tau_{h+1})$ can be computed from $\mathbf{b}(\tau_h), o_h, a_h$.

We assume that both $\pi_b$ and $\pi_e$ have the same memory; if they differ, we can simply concatenate their memories together. Then, handling memories in our analyses takes two steps:

1. We allow latent state transition to depend on $o_h$, that is, $s_{h+1} \sim \mathbb{T}(\cdot | s_h, a_h, o_h)$. This model has been considered by Jiang et al. [2017] to unify POMDPs and low-rank MDPs. **Our analyses hold as-is without any changes.** To provide some intuition: the key property that enables the analyses of FDVF is that $s_h$ is a bottleneck that separates histories from futures, which enables $(\mathcal{B}^{\mathcal{H}} V)(\tau_h) = \langle \mathbf{b}(\tau_h), \mathcal{B}_h^{\mathcal{S}} V \rangle$ in Eq.(2). This property is intact with the additional dependence of $s_h$ on $o_{h-1}$.

2. We provide a **blackbox reduction** from the memory-based case to the memoryless case. Define a new POMDP that is equivalent to the original one, where the latent state is $\tilde{s}_h = (s_h, m_h)$. The observation is $\tilde{o}_h = (o_h, m_h)$, which can be emitted from $\tilde{s}_h$ since $\tilde{s}_h$ contains $m_h$. The latent state transition is $\tilde{s}_{h+1} = (s_{h+1}, m_{h+1})$, where $s_{h+1} \sim \mathbb{T}(\cdot | s_h, a_h, o_h)$, and $m_{h+1}$ is updated from $m_h$ (contained in $\tilde{s}_h$), $a_h$, and $o_h$; this is why we need $o_h$ to participate in latent transitions. Now it suffices to perform OPE in the new POMDP. Data from the original POMDP can be converted to that of the new POMDP with $\tilde{o}_h = (o_h, m_h)$. $\pi_b$ and $\pi_e$ only need to depend on $\tilde{o}_h$ and become memoryless.

Given the reduction, if we design function classes that operate on the histories and futures of the new POMDP, the guarantees in the main text immediately hold. The final step is to translate the objects (e.g., futures and histories) and guarantees in the new POMDP back to the original POMDP for interpretability. Note that the history in the new POMDP is $\tilde{\tau}_h = (\tilde{o}_{1:h-1}, a_{1:h-1})$, which contains the same information as $\tau_h$, so functions in $\Xi$ and $\mathcal{W}$ can still operate on $\tau_h$. Similarly, $V \in \mathcal{V}$ now takes $(m_h, f_h)$ as input, which is consistent with Uehara et al. [2022a].

When translating the assumptions back to the original POMDP, we can see that now the results can be well-behaved when $m_h$ is "simple". For example, if $m_h$ takes values from a constant-sized space, $d^{\pi_e}(s_h, m_h)/d^{\pi_b}(s_h, m_h)$ (which lower-bounds belief coverage as discussed above) may not blow up exponentially, even though $m_h$ can hold information that is arbitrarily old (e.g., it remembers one bit of information from $h = 1$). This is a scenario that cannot be handled by the formulation of Uehara et al. [2022a].

However, the guarantee can still deteriorate when the policies maintain rich memories, even if these memories are highly structured, such as $m_h = \mathbf{b}(\tau_h)$. Under the mild assumption that all histories lead to distinct belief states $\mathbf{b}(\tau_h)$ (they can be very close in $\mathbb{R}^S$ and just need to be not exactly identical), the belief state in the new POMDP, $\tilde{\mathbf{b}}(\tau_h)$, completely ignores the linear structure of $m_h$ and treats it in the same way as $m_h = \tau_h$,[7] leading to an exponentially large belief coverage. How to handle policies that depend on rich but highly structured memories such as belief states is a major open problem.

---

[6]When $h \leq M$ no dropping is needed. In fact, if we never drop, then $m_h = \tau_h$ is just the history.

[7]$\tilde{\mathbf{b}}(\tau_h)$ has a "block one-hot" structure, where the block of size $S$ indexed by $m_h(\tau_h)$ is equal to $\mathbf{b}(\tau_h)$, and all other entries are 0.

## B.7 Recovering MDP Algorithms and Analyses

One somewhat undesirable property of our algorithms and analyses is that they do not subsume MDP algorithms/analyses as a special case. MDPs can be viewed as POMDPs with identity emission, i.e., $o_h = s_h$. In this case, algorithms considered in this paper are analogous to their MDP counterparts Uehara et al. [2020], except that the MDP algorithms require that all functions $V, w, \xi$ to operate on the current state $s_h$. In contrast, our FDVF operates on $f_h = (o_h, a_h, \ldots, o_H, a_H)$ which includes $o_h(= s_h)$. To recover the MDP algorithm as a special case, we can choose $\mathcal{V}$ such that every $V \in \mathcal{V}$ only depends on $f_h$ through $o_h$. However, $w$ and $\xi$ operate on $\tau_h = (o_1, a_1, \ldots, o_{h-1}, a_{h-1})$ which does not contain $o_h$. This makes subsuming the MDP case difficult.

Here we describe briefly how to overcome this issue by slightly modifying our analysis; the changes are somewhat similar to Appendix B.6. Instead of letting $\tau_h = (o_1, a_1, \ldots, o_{h-1}, a_{h-1})$, we can define an alternative notion of history $\tilde{\tau}_h = (o_1, a_1, \ldots, o_h)$ and replace $\tau_h$ in the main text with $\tilde{\tau}_h$. Algorithmically, we can immediately recover MDP algorithms by also restricting functions in $\mathcal{W}$ and $\mathcal{X}$ to only operate on $o_h$. However, our analyses (which apply to general POMDPs) need to change accordingly, as replacing $\tau_h$ with $\tilde{\tau}_h$ will break the key properties, such as $\mathcal{B}^{\mathcal{H}}V$ being linear in $\mathcal{B}^{\mathcal{S}}V$. The problem is that in the definitions of $V_{\mathcal{S}}^{\pi_e}$ and $\mathcal{B}^{\mathcal{S}}V$ we want to marginalize out the randomness of $o_h$ given $s_h$, which is in conflict with conditioning on $\tilde{\tau}_h$ that includes all the information of $o_h$. To address this, we simply replace $s_h$ with $\tilde{s}_h := (s_h, o_h)$ in the definition of $V_{\mathcal{S}}^{\pi_e}$, $\mathcal{B}^{\mathcal{S}}V$, $M_{\mathcal{H},h}$, and $M_{\mathcal{F},h}$. That is, $V_{\mathcal{S}}^{\pi_e}$ is now a function of $\tilde{s}_h$ and also depends on $o_h$, $\mathbf{b}(\tau_h)$ is the posterior distribution over $\tilde{s}_h$, and the entries of the outcome matrix $M_{\mathcal{F},h}$ is $\Pr_{\pi_b}(f_h|s_h, o_h)$. This retains the key property that $\mathcal{B}^{\mathcal{H}}V$ is linear in $\mathcal{B}^{\mathcal{S}}V$ with $\tilde{\mathbf{b}}(\tau_h)$ as the coefficient.

When we specialize the guarantees of this modified analysis to the MDP setting, outcome coverage is always satisfied and we can always use the MDP's standard value function as $V_{\mathcal{F}}$. For belief coverage, note that $(\mathcal{B}^{\mathcal{H}}V)(\tau_h) = (\mathcal{B}^{\mathcal{S}}V)(s_h, o_h)$, which is simply the standard definition of Bellman error in MDPs (since $s_h = o_h$), which we denote as $(\mathcal{B}V)(s_h)$. Plugging this into the refined coverage discussed in Appendix B.3, we recover the standard definition of coverage in the MDP setting, namely $\sup_{V \in \mathcal{V}} \frac{|\mathbb{E}_{\pi_e}[(\mathcal{B}^{\mathcal{S}}V)(s_h)]|}{\sqrt{\mathbb{E}_{\pi_b}[(\mathcal{B}^{\mathcal{S}}V)(s_h)^2]}}$.

## B.8 Incorporating Different Latent-State Priors in $V_{\mathcal{F}}$

In Section 4 we showed that $\Sigma_{\mathcal{F},h}$, which is crucial to the construction in Eq.(4), can be viewed as the confusion matrix of making posterior predictions of $s_h$ from $f_h$, using a uniform prior. The uniform prior corresponds to the all-one vector $\mathbf{1}_{\mathcal{S}}$ in the definition of $Z_h := \text{diag}(\mathbf{1}_{\mathcal{S}}^{\top}M_{\mathcal{F},h})$, which naturally leads to the question of whether we can incorporate a different and perhaps more informative prior.

Let $\mathbf{p}_h \in \Delta(\mathcal{S}_h)$ be the prior we would like to use instead of the uniform prior. An immediate idea is to define $Z_h^{\mathbf{P}_h} = \text{diag}(\mathbf{p}_h^{\top}M_{\mathcal{F},h})$ (possibly up to a $S$ scaling factor, as the uniform prior is $\mathbf{p}_{\text{unif}} = [1/S, \cdots, 1/S]^{\top}$ and $\mathbf{1}_{\mathcal{S}} = S\mathbf{p}_{\text{unif}}$) and directly plug it into the construction in Eq.(4). However, this breaks some of the key properties of the current construction of $\Sigma_{\mathcal{F},h}$, such as $L_1$ normalization of rows of $Z_h^{-1}M_{\mathcal{F},h}^{\top}$.

To resolve this, the key is to realize that the rows of $Z_h^{-1}M_{\mathcal{F},h}^{\top}$ are $L_1$ normalized because they can be viewed as posteriors over $\mathcal{S}_h$. In comparison, if we examine the $(f_h, s_h)$-th entry of $(Z_h^{\mathbf{P}_h})^{-1}M_{\mathcal{F},h}^{\top}$, it is

$$\frac{\Pr_{\pi_b}[f_h|s_h]}{\sum_{s' \in \mathcal{S}_h} \Pr_{\pi_b}[f_h|s']\mathbf{p}_h(s')},$$

so clearly it is missing a $\mathbf{p}_h(s_h)$ term on the numerator to be a proper posterior. Inspired by this observation, we can see how to fix the construction now: recall that $V_{\mathcal{F}}$ needs to satisfy $M_{\mathcal{F},h}V_{\mathcal{F},h} = V_{\mathcal{S},h}^{\pi_e}$. If $\mathbf{p}_h > 0$, then this is equivalent to

$$(\text{diag}(\mathbf{p}_h)M_{\mathcal{F},h})V_{\mathcal{F},h} = \text{diag}(\mathbf{p}_h)V_{\mathcal{S},h}^{\pi_e}.$$

Then, the minimum $Z_h^{\mathbf{P}_h}$-weighted solution of this equation will provide the desired construction that preserves the $L_1$ normalization properties.

**Which prior $\mathbf{p}_h$ to use?** For the initial time step $h = 1$, the initial latent-state distribution $d_1$ is the most natural candidate for the prior. If $d_1(s_1) = 0$ for some $s_1 \in \mathcal{S}_1$, incorporating $d_1$ as the

prior will essentially treat $s_1$ as non-existent and ignore the outcome coverage of $\pi_b$ over $\pi_e$ from $s_1$, which is reasonable because $s_1$ will not be activated by neither $\pi_e$ and $\pi_b$. For $h > 1$, the answer is less clear, and natural candidates include $\mathbf{p}_h = d_h^{\pi_b}$ and $d_h^{\pi_e}$, or perhaps their probability mixture. For the example scenarios examined in the main text, these choices (or even a uniform prior) do not make significant differences, and we leave the investigation of which prior is the best to future work.

## C  Proofs for Section 3

### C.1  Proof of Lemma 1

The RHS of the lemma statement is

$$\sum_{h=1}^{H} \mathbb{E}_{\pi_e} \left[ (\mathcal{B}^{\mathcal{S}} V)(s_h) \right] = \sum_{h=1}^{H} \mathbb{E}_{s_h \sim d_h^{\pi_e}} \left[ \mathbb{E}_{\substack{a_h \sim \pi_e \\ a_{h+1:H} \sim \pi_b}} [r_h + V(f_{h+1}) \mid s_h] - \mathbb{E}_{\pi_b} [V(f_h) \mid s_h] \right].$$

Now notice that the expected value of the $V(f_{h+1})$ term for $h$ is the same as that of the $V(f_h)$ term for $h + 1$, since both can be written as $\mathbb{E}_{s_{h+1} \sim d_{h+1}^{\pi_e}} [\mathbb{E}_{\pi_b} [V(f_{h+1}) | s_{h+1}]]$. After telescoping cancellations, what remains on the RHS is

$$\sum_{h=1}^{H} \mathbb{E}_{s_h \sim d_h^{\pi_e}} \left[ \mathbb{E}_{\substack{a_h \sim \pi_e \\ a_{h+1:H} \sim \pi_b}} [r_h \mid s_h] \right] - \mathbb{E}_{s_1} [\mathbb{E}_{\pi_b} [V(f_1)|s_1]] = J(\pi_e) - \mathbb{E}_{\pi_b} [V(f_1)]. \qquad \square$$

### C.2  Proof of Theorem 2

The proof follows similar idea from Uehara et al. [2022a]. Let $\mathcal{G}_h V = \mu(o_h, a_h)(r_h + V(f_{h+1})) - V(f_h)$, we have

$$\widehat{V} = \operatorname*{argmin}_{V \in \mathcal{V}} \max_{\xi \in \Xi} \sum_{h=1}^{H} \mathbb{E}_{\mathcal{D}} \left[ (\mathcal{G}_h V)^2 - (\mathcal{G}_h V - \xi(\tau_h))^2 \right].$$

For any fixed $V$, we define

$$\widehat{\xi}_V = \operatorname*{argmax}_{\xi \in \Xi} - \sum_{h=1}^{H} \mathbb{E}_{\mathcal{D}} \left[ (\mathcal{G}_h V - \xi(\tau_h))^2 \right].$$

**Analysis of Inner Maximizer.**  We observe that for any $h \in [H]$,

$$\mathbb{E}_{\pi_b} \left[ (\xi(\tau_h) - \mathcal{G}_h V)^2 - ((\mathcal{B}^{\mathcal{H}} V)(\tau_h) - \mathcal{G}_h V)^2 \right] = \mathbb{E}_{\pi_b} \left[ (\xi(\tau_h) - (\mathcal{B}^{\mathcal{H}} V)(\tau_h))^2 \right].$$

Let $X_h = (\xi(\tau_h) - \mathcal{G}_h V)^2 - ((\mathcal{B}^{\mathcal{H}} V)(\tau_h) - \mathcal{G}_h V)^2$ and $X = \sum_{h=1}^{H} X_h$. Let $\bar{C} = \max\{1 + C_{\mathcal{V}}, C_{\Xi}\}$, since $C_\mu \geq 1$, we have $|\xi(\tau_h)| \leq \bar{C}$, $|\mathcal{G}_h V| \leq 3C_\mu \bar{C}$ and $|\mathcal{B}^{\mathcal{H}} V(\tau_h)| \leq 3\bar{C}$. Therefore, we have $|X_h| \leq 40 C_\mu \bar{C}^2$ and $|X| \leq 40 H C_\mu \bar{C}^2$. We observe that

$$\mathbb{E}_{\pi_b}[X_h^2]$$
$$\leq \mathbb{E}_{\pi_b} \left[ ((\mathcal{B}^{\mathcal{H}} V)(\tau_h) - \xi(\tau_h))^2 \left( (\mathcal{B}^{\mathcal{H}} V)(\tau_h) + \xi(\tau_h) - 2\mathcal{G}_h V \right)^2 \right]$$
$$\leq \mathbb{E}_{\pi_b} \left[ ((\mathcal{B}^{\mathcal{H}} V)(\tau_h) - \xi(\tau_h))^2 \left( 30\bar{C}^2 + 12(\mathcal{G}_h V)^2 \right) \right]$$
$$\leq \mathbb{E}_{\pi_b} \left[ ((\mathcal{B}^{\mathcal{H}} V)(\tau_h) - \xi(\tau_h))^2 \left( 54\bar{C}^2 + 96\bar{C}^2 \mu(o_h, a_h)^2 \right) \right]$$
$$\leq 150 \bar{C}^2 C_\mu \mathbb{E}_{\pi_b} \left[ ((\mathcal{B}^{\mathcal{H}} V)(\tau_h) - \xi(\tau_h))^2 \right].$$

Hence, for the variance of $X$, we have

$$\operatorname{Var}_{\pi_b}[X] \leq H \sum_{h=1}^{H} \mathbb{E}_{\pi_b} \left[ X_h^2 \right]$$
$$\leq 150 H \bar{C}^2 C_\mu \sum_{h=1}^{H} \mathbb{E}_{\pi_b} \left[ ((\mathcal{B}^{\mathcal{H}} V)(\tau_h) - \xi(\tau_h))^2 \right].$$

From Bernstein's inequality, with probability at least $1 - \delta/2$, $\forall V \in \mathcal{V}$, $\forall \xi \in \Xi$, we have

$$\left| \sum_{h=1}^{H} \{\mathbb{E}_{\mathcal{D}} - \mathbb{E}_{\pi_b}\}[(\xi(\tau_h) - \mathcal{G}_h V)^2 - ((\mathcal{B}^{\mathcal{H}} V)(\tau_h) - \mathcal{G}_h V)^2] \right|$$

$$\leq \sqrt{\frac{150 H \bar{C}^2 C_\mu \sum_{h=1}^{H} \mathbb{E}_{\pi_b}[((\mathcal{B}^{\mathcal{H}} V)(\tau_h) - \xi(\tau_h))^2] \log(4|\mathcal{V}||\Xi|/\delta)}{n}} + \frac{40 H \bar{C}^2 C_\mu \log(4|\mathcal{V}||\Xi|/\delta)}{n}.$$

$$(11)$$

From Bellman completeness assumption $\mathcal{B}^{\mathcal{H}} \mathcal{V} \subset \Xi$, we also have

$$\sum_{h=1}^{H} \mathbb{E}_{\mathcal{D}}[(\widehat{\xi}_V - \mathcal{G}_h V)^2 - ((\mathcal{B}^{\mathcal{H}} V)(\tau_h) - \mathcal{G}_h V)^2] \leq 0. \tag{12}$$

Therefore, we obtain that

$$\sum_{h=1}^{H} \mathbb{E}_{\pi_b}[((\mathcal{B}^{\mathcal{H}} V)(\tau_h) - \widehat{\xi}_V(\tau_h))^2]$$

$$= \sum_{h=1}^{H} \mathbb{E}_{\pi_b}[(\widehat{\xi}_V(\tau_h) - \mathcal{G}_h V)^2 - ((\mathcal{B}^{\mathcal{H}} V)(\tau_h) - \mathcal{G}_h V)^2]$$

$$\leq \left| \sum_{h=1}^{H} \{\mathbb{E}_{\mathcal{D}} - \mathbb{E}_{\pi_b}\}[(\widehat{\xi}_V(\tau_h) - \mathcal{G}_h V)^2 - ((\mathcal{B}^{\mathcal{H}} V)(\tau_h) - \mathcal{G}_h V)^2] \right| + \sum_{h=1}^{H} \mathbb{E}_{\mathcal{D}}[(\widehat{\xi}_V(\tau_h) - \mathcal{G}_h V)^2 - ((\mathcal{B}^{\mathcal{H}} V)(\tau_h) - \mathcal{G}_h V)^2]$$

$$\leq \left| \sum_{h=1}^{H} \{\mathbb{E}_{\mathcal{D}} - \mathbb{E}_{\pi_b}\}[(\widehat{\xi}_V(\tau_h) - \mathcal{G}_h V)^2 - ((\mathcal{B}^{\mathcal{H}} V)(\tau_h) - \mathcal{G}_h V)^2] \right|$$

$$\leq \sqrt{\frac{150 H \bar{C}^2 C_\mu \sum_{h=1}^{H} \mathbb{E}_{\pi_b}[((\mathcal{B}^{\mathcal{H}} V)(\tau_h) - \widehat{\xi}_V(\tau_h))^2] \log(4|\mathcal{V}||\Xi|/\delta)}{n}} + \frac{40 H \bar{C}^2 C_\mu \log(4|\mathcal{V}||\Xi|/\delta)}{n}.$$

The second inequality is from Equation (12) and the last inequality is from Equation (11). We then have

$$\sum_{h=1}^{H} \mathbb{E}_{\pi_b}[((\mathcal{B}^{\mathcal{H}} V)(\tau_h) - \widehat{\xi}_V(\tau_h))^2] \leq \varepsilon_{\text{stat}}, \quad \varepsilon_{\text{stat}} := \frac{225 H \bar{C}^2 C_\mu \log(4|\mathcal{V}||\Xi|/\delta)}{n}. \tag{13}$$

Combining Equation (11) and Equation (13), we have

$$\left| \sum_{h=1}^{H} \{\mathbb{E}_{\mathcal{D}} - \mathbb{E}_{\pi_b}\}[(\widehat{\xi}_V(\tau_h) - \mathcal{G}_h V)^2 - ((\mathcal{B}^{\mathcal{H}} V)(\tau_h) - \mathcal{G}_h V)^2] \right| \leq \varepsilon_{\text{stat}}. \tag{14}$$

Hence,

$$\left| \sum_{h=1}^{H} \mathbb{E}_{\mathcal{D}}[(\widehat{\xi}_V(\tau_h) - \mathcal{G}_h V)^2] - \sum_{h=1}^{H} \mathbb{E}_{\mathcal{D}}[((\mathcal{B}^{\mathcal{H}} V)(\tau_h) - \mathcal{G}_h V)^2] \right|$$

$$\leq \left| \sum_{h=1}^{H} \mathbb{E}_{\pi_b}[(\widehat{\xi}_V(\tau_h) - \mathcal{G}_h V)^2] - \sum_{h=1}^{H} \mathbb{E}_{\pi_b}[((\mathcal{B}^{\mathcal{H}} V)(\tau_h) - \mathcal{G}_h V)^2] \right| + 2\varepsilon_{\text{stat}}$$

$$= \left| \sum_{h=1}^{H} \mathbb{E}_{\pi_b}[((\mathcal{B}^{\mathcal{H}} V)(\tau_h) - \widehat{\xi}_V(\tau_h))^2] \right| + 2\varepsilon_{\text{stat}} \leq 3\varepsilon_{\text{stat}}.$$

The first inequality is from Equation (14) and the last step is from Equation (13).

**Analysis of Outer Minimizer.** For any future-dependent value function $V_{\mathcal{F}}$, from optimality of $\widehat{V}$ and the convergence of inner maximizer, we have

$$\sum_{h=1}^{H} \mathbb{E}_{\mathcal{D}}[(\mathcal{G}_h\widehat{V})^2 - (\mathcal{G}_h\widehat{V} - ((\mathcal{B}^{\mathcal{H}}\widehat{V}))(\tau_h))^2] \leq \sum_{h=1}^{H} \mathbb{E}_{\mathcal{D}}\left[(\mathcal{G}_h\widehat{V})^2 - (\mathcal{G}_h\widehat{V} - \widehat{\xi}_{\widehat{V}})^2\right] + 3\varepsilon_{\text{stat}}.$$

$$\leq \sum_{h=1}^{H} \mathbb{E}_{\mathcal{D}}\left[(\mathcal{G}_hV_{\mathcal{F}})^2 - (\mathcal{G}_hV_{\mathcal{F}} - \widehat{\xi}_{V_{\mathcal{F}}})^2\right] + 3\varepsilon_{\text{stat}}$$

$$\leq \sum_{h=1}^{H} \mathbb{E}_{\mathcal{D}}\left[(\mathcal{G}_hV_{\mathcal{F}})^2 - (\mathcal{G}_hV_{\mathcal{F}} - (\mathcal{B}^{\mathcal{H}}V_{\mathcal{F}})(\tau_h))^2\right] + 6\varepsilon_{\text{stat}}$$

$$= 6\varepsilon_{\text{stat}}. \tag{15}$$

The last step is from that $(\mathcal{B}^{\mathcal{H}}V_{\mathcal{F}})(\tau_h) = 0$. For any $\tau_h$, we observe that $\forall V \in \mathcal{V}$ and $h \in [H]$,

$$\mathbb{E}_{\pi_b}[(\mathcal{G}_hV)^2 - (\mathcal{G}_hV - (\mathcal{B}^{\mathcal{H}}V)(\tau_h))^2]$$
$$= \mathbb{E}_{\pi_b}[-(\mathcal{B}^{\mathcal{H}}V)(\tau_h)^2 + 2(\mathcal{B}^{\mathcal{H}}V)(\tau_h)\mathcal{G}_hV]$$
$$= \mathbb{E}_{\pi_b}[(\mathcal{B}^{\mathcal{H}}V)(\tau_h)^2].$$

For any fixed $V \in \mathcal{V}$, let $Y_h = (\mathcal{G}_hV)^2 - (\mathcal{G}_hV - (\mathcal{B}^{\mathcal{H}}V)(\tau_h))^2$ and $Y = \sum_{h=1}^{H} Y_h$, we have $|Y_h| \leq 27\bar{C}^2C_\mu$ and $|Y| \leq 27H\bar{C}^2C_\mu$. We observe that

$$\mathbb{E}_{\pi_b}[Y_h^2] = \mathbb{E}_{\pi_b}[(2\mathcal{G}_hV - (\mathcal{B}^{\mathcal{H}}V)(\tau_h))^2(\mathcal{B}^{\mathcal{H}}V)(\tau_h)^2]$$
$$\leq \mathbb{E}_{\pi_b}[(18\bar{C}^2 + 4(\mathcal{G}_hV)^2)(\mathcal{B}^{\mathcal{H}}V)(\tau_h)^2]$$
$$\leq \mathbb{E}_{\pi_b}[(26\bar{C}^2 + 32\bar{C}^2\mu(o_h, a_h)^2)(\mathcal{B}^{\mathcal{H}}V)(\tau_h)^2]$$
$$\leq 58\bar{C}^2C_\mu\mathbb{E}_{\pi_b}[(\mathcal{B}^{\mathcal{H}}V)(\tau_h)^2].$$

Then, for the variance of $Y$, we have

$$\text{Var}_{\pi_b}[Y] \leq \mathbb{E}_{\pi_b}[Y^2]$$
$$\leq H\sum_{h=1}^{H} \mathbb{E}_{\pi_b}[Y_h^2]$$
$$= 58H\bar{C}^2C_\mu\sum_{h=1}^{H} \mathbb{E}_{\pi_b}[(\mathcal{B}^{\mathcal{H}}V)(\tau_h)^2].$$

From Bernstein's inequality, with probability at least $1 - \delta/2$, $\forall V \in \mathcal{V}$, we have

$$\left|\sum_{h=1}^{H}(\mathbb{E}_{\mathcal{D}} - \mathbb{E}_{\pi_b})\left[(\mathcal{G}_hV)^2 - (\mathcal{G}_hV - (\mathcal{B}^{\mathcal{H}}V)(\tau_h))^2\right]\right|$$
$$\leq \sqrt{58H\bar{C}^2C_\mu\sum_{h=1}^{H} \mathbb{E}_{\pi_b}[(\mathcal{B}^{\mathcal{H}}V)(\tau_h)^2]\frac{\log(4|\mathcal{V}|/\delta)}{n}} + \frac{27H\bar{C}^2C_\mu\log(4|\mathcal{V}|/\delta)}{n}. \tag{16}$$

Therefore, we have

$$\sum_{h=1}^{H} \mathbb{E}_{\pi_b} \left[ (\mathcal{B}^{\mathcal{H}} \widehat{V})(\tau_h)^2 \right] = \sum_{h=1}^{H} \mathbb{E}_{\pi_b} [(\mathcal{G}_h \widehat{V})^2 - (\mathcal{G}_h \widehat{V} - (\mathcal{B}^{\mathcal{H}} \widehat{V})(\tau_h))^2]$$

$$\leq \left| \sum_{h=1}^{H} (\mathbb{E}_{\mathcal{D}} - \mathbb{E}_{\pi_b})[(\mathcal{G}_h \widehat{V})^2 - (\mathcal{G}_h \widehat{V} - (\mathcal{B}^{\mathcal{H}} \widehat{V})(\tau_h))^2] \right|$$

$$+ \sum_{h=1}^{H} \mathbb{E}_{\mathcal{D}} \left[ (\mathcal{G}_h \widehat{V})^2 - (\mathcal{G}_h \widehat{V} - (\mathcal{B}^{\mathcal{H}} \widehat{V})(\tau_h))^2 \right]$$

$$\leq \sqrt{58 H \bar{C}^2 C_\mu \sum_{h=1}^{H} \mathbb{E}_{\pi_b} \left[ (\mathcal{B}^{\mathcal{H}} \widehat{V})(\tau_h)^2 \right] \frac{\log(4|\mathcal{V}|/\delta)}{n}} + \frac{27 H \bar{C}^2 C_\mu \log(4|\mathcal{V}|/\delta)}{n} + 6\varepsilon_{\text{stat}}.$$

(From Equation (15) and Equation (16))

Solving it and we get

$$\sum_{h=1}^{H} \mathbb{E}_{\pi_b} \left[ (\mathcal{B}^{\mathcal{H}} \widehat{V})(\tau_h)^2 \right] \leq 10\varepsilon_{\text{stat}}.$$

We then invoke Lemma 1 and obtain that

$$\left| J(\pi_e) - \mathbb{E}_{\pi_b}[\widehat{V}(f_1)] \right| \leq \left| \sum_{h=1}^{H} \mathbb{E}_{s_h \sim d^h_{\pi_e}} \left[ (\mathcal{B}^{\mathcal{S}} \widehat{V})(s_h) \right] \right|$$

$$\leq \sqrt{H \sum_{h=1}^{H} \left( \mathbb{E}_{s_h \sim d^h_{\pi_e}} \left[ (\mathcal{B}^{\mathcal{S}} \widehat{V})(s_h) \right] \right)^2}$$

$$\leq \sqrt{H \sum_{h=1}^{H} \mathbb{E}_{s_h \sim d^h_{\pi_e}} \left[ (\mathcal{B}^{\mathcal{S}} \widehat{V})(s_h)^2 \right]}$$

$$\leq \sqrt{H \sum_{h=1}^{H} \mathbb{E}_{\pi_b} \left[ (\mathcal{B}^{\mathcal{H}} \widehat{V})(\tau_h)^2 \right] \cdot \sqrt{\frac{\sum_{h=1}^{H} \mathbb{E}_{s_h \sim d^h_{\pi_e}} \left[ (\mathcal{B}^{\mathcal{S}} \widehat{V})(s_h)^2 \right]}{\sum_{h=1}^{H} \mathbb{E}_{\pi_b} \left[ (\mathcal{B}^{\mathcal{H}} \widehat{V})(\tau_h)^2 \right]}}}$$

$$\leq \sqrt{10 H \varepsilon_{\text{stat}}} \cdot \sqrt{\frac{\sum_{h=1}^{H} \mathbb{E}_{\pi_e} \left[ (\mathcal{B}^{\mathcal{S}} \widehat{V})(s_h)^2 \right]}{\sum_{h=1}^{H} \mathbb{E}_{\pi_b} \left[ (\mathcal{B}^{\mathcal{S}} \widehat{V})(s_h)^2 \right]}} \cdot \sqrt{\frac{\sum_{h=1}^{H} \mathbb{E}_{\pi_b} \left[ (\mathcal{B}^{\mathcal{S}} \widehat{V})(s_h)^2 \right]}{\sum_{h=1}^{H} \mathbb{E}_{\pi_b} \left[ (\mathcal{B}^{\mathcal{H}} \widehat{V})(\tau_h)^2 \right]}}$$

$$\leq 50 H \max\{C_{\mathcal{V}} + 1, C_{\Xi}\} \text{IV}(\mathcal{V}) \text{Dr}_{\mathcal{V}}[d^{\pi_e}, d^{\pi_b}] \sqrt{\frac{C_\mu \log \frac{4|\mathcal{V}||\Xi|}{\delta}}{n}}.$$

The proof is completed by using Hoeffding's inequality to bound $|\mathbb{E}_{\mathcal{D}}[\widehat{V}(f_1)] - \mathbb{E}_{\pi_b}[\widehat{V}(f_1)]|$. $\qquad\square$

## D   Proofs for Section 4

### D.1   Example 1

We provide a brief justification of Example 1. When $\Pr_{\pi_b}(f_h|s_h) \leq \frac{C_{\text{stoch}}}{(OA)^{H-h+1}}$, for any $\mathbf{u}(f_h)$, we have

$$\|\mathbf{u}(f_h)\|_2 \leq \frac{C_{\text{stoch}} \sqrt{S}}{(OA)^{H-h+1}}.$$

Then, for any $\mathbf{x} \in \mathbb{R}^S$ such that $\|\mathbf{x}\|_2 = 1$, we have

$$\mathbf{x}^\top (M_{\mathcal{F},h} M_{\mathcal{F},h}^\top) \mathbf{x} \leq \sum_{f_h} \|\mathbf{x}\|_2^2 \|\mathbf{u}(f_h)\|_2^2 = \frac{C_{\text{stoch}}^2 S}{(OA)^{H-h+1}}.$$

Therefore, $\sigma_{\min}(M_{\mathcal{F},h}) \leq \sigma_{\max}(M_{\mathcal{F},h}) \leq C_{\text{stoch}}\sqrt{S}/(OA)^{(H-h+1)/2}$.

## D.2   Proof of Proposition 3

For each entry in $\Sigma_{\mathcal{F},h}$, we can write it as

$$(\Sigma_{\mathcal{F},h})_{ij} = \sum_k \frac{\Pr_{\pi_b}(f_h = k \mid s_h = i)\Pr_{\pi_b}(f_h = k \mid s_h = j)}{\sum_{i'}\Pr_{\pi_b}(f_h = k \mid s_h = i')}$$

Since the probability is always non-negative, all the entries in $\Sigma_{\mathcal{F},h}$ is non-negative. For each row $i$, we have

$$\sum_j (\Sigma_{\mathcal{F},h})_{ij} = \sum_k \sum_j \frac{\Pr_{\pi_b}(f_h = k \mid s_h = i)\Pr_{\pi_b}(f_h = k \mid s_h = j)}{\sum_{i'}\Pr_{\pi_b}(f_h = k \mid s_h = i')}$$
$$= \sum_k \Pr_{\pi_b}(f_h = k \mid s_h = i) = 1.$$

Similarly, for each column $j$, we also have $\sum_i (\Sigma_{\mathcal{F},h})_{ij} = 1$. Therefore, we prove that $\Sigma_{\mathcal{F},h}$ is doubly-stochastic and we know for a stochastic matrix, the largest eigenvalue is 1.   □

## D.3   Example 2

Due to the block structure of $M_{\mathcal{F},h}$, $\Sigma_{\mathcal{F},h}$ is clearly diagonal, and the $i$-th diagonal entry is:

$$\sum_{j:\Pr_{\pi_b}[f_h=j|s_h=i]>0} \frac{\Pr_{\pi_b}[f_h = j|s_h = i]^2}{\sum_{i'}\Pr_{\pi_b}[f_h = j|s_h = i']}.$$

Due to the revealing property, the denominator is just $\Pr_{\pi_b}[f_h = j|s_h = i]$, so the expression is just summing up $\Pr_{\pi_b}[f_h = j|s_h = i]$ over $j$, which is 1.

## D.4   Proof of Proposition 4

According to Equation (4), we have for any $f_h$,

$$V_{\mathcal{F}}(f_h) = Z(f_h)^{-1}\mathbf{u}(f_h)^\top \Sigma_{\mathcal{F},h}^{-1} V_{\mathcal{S},h}^{\pi_e}$$
$$\leq \sqrt{Z(f_h)^{-1}\mathbf{u}(f_h)^\top \Sigma_{\mathcal{F},h}^{-1} Z(f_h)^{-1}\mathbf{u}(f_h)}\sqrt{(V_{\mathcal{S},h}^{\pi_e})^\top \Sigma_{\mathcal{F},h}^{-1} V_{\mathcal{S},h}^{\pi_e}}$$
$$\leq \sqrt{C_{\mathcal{F},U}C_{\mathcal{F},V}}.$$

The last step is from Assumption 7 and Assumption 8. Therefore, $\|V_{\mathcal{F}}\|_\infty \leq \sqrt{C_{\mathcal{F},U}C_{\mathcal{F},V}} = \sqrt{C_{\mathcal{F},2}}$. Moreover, we observe that

$$\|V_{\mathcal{F},h}\|_{Z_h}^2 = \sum_{f_h} Z(f_h)\left(Z(f_h)^{-1}\mathbf{u}(f_h)^\top \Sigma_{\mathcal{F},h}^{-1} V_{\mathcal{S}}^{\pi_e}\right)^2$$
$$= \sum_{f_h} Z(f_h)^{-1}(V_{\mathcal{S}}^{\pi_e})^\top \Sigma_{\mathcal{F},h}^{-1}\mathbf{u}(f_h)\mathbf{u}(f_h)^\top \Sigma_{\mathcal{F},h}^{-1} V_{\mathcal{S}}^{\pi_e}$$
$$= (V_{\mathcal{S}}^{\pi_e})^\top \Sigma_{\mathcal{F},h}^{-1}\left(\sum_{f_h} Z(f_h)^{-1}\mathbf{u}(f_h)\mathbf{u}(f_h)^\top\right)\Sigma_{\mathcal{F},h}^{-1} V_{\mathcal{S}}^{\pi_e}$$
$$= (V_{\mathcal{S}}^{\pi_e})^\top \Sigma_{\mathcal{F},h}^{-1}\Sigma_{\mathcal{F},h}\Sigma_{\mathcal{F},h}^{-1} V_{\mathcal{S}}^{\pi_e} \leq C_{\mathcal{F},V}.$$

The last step is from Assumption 7.   □

## D.5 Example 3

We give the calculation for Example 3. When $\pi_e = \pi_b$, $V^{\pi_e}_{\mathcal{S},h} = M_{\mathcal{F},h} R^+_h$, so

$$
\begin{aligned}
\|V^{\pi_e}_{\mathcal{S},h}\|^2_{\Sigma^{-1}_{\mathcal{F},h}} &= (R^+_h)^\top M^\top_{\mathcal{F},h} \Sigma^{-1}_{\mathcal{F},h} M_{\mathcal{F},h} R^+_h \\
&= (R^+_h)^\top Z^{1/2}_h Z^{-1/2}_h M^\top_{\mathcal{F},h} (M_{\mathcal{F},h} Z^{-1}_h M^\top_{\mathcal{F},h}) M_{\mathcal{F},h} Z^{-1/2}_h Z^{1/2}_h R^+_h \\
&\leq (R^+_h)^\top Z^{1/2}_h Z^{1/2}_h R^+_h,
\end{aligned}
$$

where the last step follows from the fact that $Z^{-1/2}_h M^\top_{\mathcal{F},h} (M_{\mathcal{F},h} Z^{-1}_h M^\top_{\mathcal{F},h}) M_{\mathcal{F},h} Z^{-1/2}_h$ is a projection matrix ($P^2 = P$) and is dominated by identity in eigenvalues ($P \preceq I$). Now, recall that $Z_h/S$ is a proper distribution, so

$$
(R^+_h)^\top Z_h R^+_h = S \cdot \mathbb{E}_{Z_h/S}[(R^+_h)^2] \leq SH^2.
$$

# E   Proofs for Section 5

## E.1   Proof of Lemma 6

We first verify that $w^\star$ in Eq. (8) satisfies Eq. (7) as follows.

$$
\begin{aligned}
\mathbb{E}_{\pi_b}[w^\star(\tau_h)\mathbf{b}(\tau_h)] &= \sum_{\tau_h} d^{\pi_b}_h(\tau_h) w^\star(\tau_h)\mathbf{b}(\tau_h) \\
&= \sum_{\tau_h} d^{\pi_b}_h(\tau_h)\mathbf{b}(\tau_h)\mathbf{b}(\tau_h)^\top \Sigma^{-1}_{\mathcal{H},h}\mathbf{b}^{\pi_e}_h \\
&= \left( \sum_{\tau_h} d^{\pi_b}_h(\tau_h)\mathbf{b}(\tau_h)\mathbf{b}(\tau_h)^\top \right) \Sigma^{-1}_{\mathcal{H},h}\mathbf{b}^{\pi_e}_h \\
&= \mathbf{b}^{\pi_e}_h.
\end{aligned}
$$

We then show that

$$
\begin{aligned}
\|w^\star\|^2_{2,d^{\pi_b}_h} &= \sum_{\tau_h} d^{\pi_b}_h(\tau_h) \left\{ \mathbf{b}(\tau_h)^\top \Sigma^{-1}_{\mathcal{H},h}(\mathbf{b}^{\pi_e}_h) \right\}^2 \\
&= (\mathbf{b}^{\pi_e}_h)^\top \Sigma^{-1}_{\mathcal{H},h} \left( \sum_{\tau_h} d^{\pi_b}_h(\tau_h)\mathbf{b}(\tau_h)\mathbf{b}(\tau_h)^\top \right) \Sigma^{-1}_{\mathcal{H},h}\mathbf{b}^{\pi_e}_h \\
&= (\mathbf{b}^{\pi_e}_h)^\top \Sigma^{-1}_{\mathcal{H},h}\Sigma_{\mathcal{H},h}\Sigma^{-1}_{\mathcal{H},h}\mathbf{b}^{\pi_e}_h \\
&= (\mathbf{b}^{\pi_e}_h)^\top \Sigma^{-1}_{\mathcal{H},h}\mathbf{b}^{\pi_e}_h \leq C_{\mathcal{H},2}.
\end{aligned}
$$

The last step follows from Assumption 11.   □

## E.2   Example 5

Consider the lemma: given vector $x$ and PD matrix $\Sigma$, if $\Sigma \succeq xx^\top$, then $x^\top \Sigma^{-1} x \leq 1$. The calculation in the example directly follows from letting $x = \mathbf{b}^{\pi_b}_h$, $\Sigma = \Sigma_{\mathcal{H},h}$, and the condition $\Sigma \succeq xx^\top$ is satisfied due to Jensen's inequality.

To prove the lemma, note that $\Sigma - xx^\top$ is PSD, so

$$
(\Sigma^{-1}x)^\top (\Sigma - xx^\top)(\Sigma^{-1}x) \geq 0.
$$

This implies $x^\top \Sigma^{-1} x \geq x^\top \Sigma^{-1} xx^\top \Sigma^{-1} x$, so $x^\top \Sigma^{-1} x \leq 1$.

## E.3   Comparison between belief coverage and latent state coverage

Here we show that belief coverage is stronger than latent state coverage. More concretely, consider a standard measure of latent state coverage, the 2nd moment of state density ratio (c.f. discussion below Example 7):

$$
\mathbb{E}_{\pi_b}[(d^{\pi_e}(s_h)/d^{\pi_b}(s_h))^2] = (\mathbf{b}^{\pi_e}_h)^\top \mathrm{diag}(\mathbb{E}_{\pi_b}[\mathbf{b}(\tau_h)])^{-1}\mathbf{b}^{\pi_e}_h.
$$

In comparison, our belief coverage parameter from Assumption 11 is

$$(\mathbf{b}_h^{\pi_e})^\top \Sigma_{\mathcal{H},h}^{-1} \mathbf{b}_h^{\pi_e} = (\mathbf{b}_h^{\pi_e})^\top \mathbb{E}_{\pi_b}[\mathbf{b}(\tau_h)\mathbf{b}(\tau_h)^\top]^{-1}\mathbf{b}_h^{\pi_e}.$$

To show the former is smaller than the latter, it suffices to show that

$$\text{diag}(\mathbb{E}_{\pi_b}[\mathbf{b}(\tau_h)])^{-1} \preceq (\mathbb{E}_{\pi_b}[\mathbf{b}(\tau_h)\mathbf{b}(\tau_h)^\top])^{-1},$$

which is implied by $\text{diag}(\mathbb{E}_{\pi_b}[\mathbf{b}(\tau_h)]) \succeq \mathbb{E}_{\pi_b}[\mathbf{b}(\tau_h)\mathbf{b}(\tau_h)^\top]$. It therefore suffices to show that $\text{diag}(\mathbf{b}(\tau_h)) \succeq \mathbf{b}(\tau_h)\mathbf{b}(\tau_h)^\top$ holds in a pointwise manner for all $\tau_h$. To show this, we temporarily let $\mathbf{b} = \mathbf{b}(\tau_h)$ in the calculation: consider an arbitrary vector $v \in \mathbb{R}^S$, then

$$v^\top (\text{diag}(\mathbf{b}) - \mathbf{b}\mathbf{b}^\top)v = \mathbb{E}_{s\sim\mathbf{b}}[v(s)^2] - (\mathbb{E}_{s\sim\mathbf{b}}[v(s)])^2 \geq 0.$$

The last step is due to Jensen's inequality.

### E.4 Comparison to IV($\mathcal{V}$)

We now compare to the IV($\mathcal{V}$) and Dr$_\mathcal{V}$ terms in Theorem 2. Belief coverage is generally stronger than latent state coverage which corresponds to the Dr$_\mathcal{V}$ term (see Appendix E.3). That said, perhaps surprisingly, the remaining IV($\mathcal{V}$) term *must* scale with $\frac{1}{\sigma_{\min}(\Sigma_{\mathcal{H},h})}$ under moderate assumptions.

**Proposition 10.** *Suppose $\mathbf{v}_{\min}$ is the eigenvector corresponding to $\sigma_{\min}(\Sigma_{\mathcal{H},h})$, the smallest eigenvalue for some $\Sigma_{\mathcal{H},h}$. Then, if $c_0\mathbf{v}_{\min} \in \mathcal{B}_h^{\mathcal{S}}\mathcal{V} := \{\mathcal{B}_h^{\mathcal{S}}V : V \in \mathcal{V}\}$ for some non-zero $c_0$,*
$$\text{IV}(\mathcal{V}) \geq \sqrt{\frac{\min_{s_h} d^{\pi_b}(s_h)}{\sigma_{\min}(\Sigma_{\mathcal{H},h})}}.$$

The proposition states that as long as $\min_{s_h} d^{\pi_b}(s_h)$ is bounded away from $0$ (which is benign and helps bound the Dr$_\mathcal{V}[d^{\pi_e}, d^{\pi_b}]$ term) and $\mathcal{V}$ is sufficiently rich that $\mathcal{B}_h^{\mathcal{S}}\mathcal{V}$ includes a certain vector in $\mathbb{R}^S$, then boundedness of IV($\mathcal{V}$) *requires* bounded $1/\sigma_{\min}(\Sigma_{\mathcal{H},h})$, which is stronger than our belief coverage assumption. This renders the split between IV($\mathcal{V}$) and Dr$_\mathcal{V}[d^{\pi_e}, d^{\pi_b}]$ superficial and perhaps unnecessary.

*Proof of Proposition 10.* Since $c_0\mathbf{v}_{\min} \in \{\mathcal{B}_h^{\mathcal{S}}V : V \in \mathcal{V}\}$, there exists $V$ such that $\mathcal{B}_h^{\mathcal{S}}V = c_0\mathbf{v}_{\min}$. For such $V$, we observe that

$$\mathbb{E}_{\pi_b}\left[(\mathcal{B}^{\mathcal{H}}V)(\tau_h)^2\right] = \sum_{\tau_h} d^{\pi_b}(\tau_h)\langle \mathbf{b}(\tau_h), c_0\mathbf{v}_{\min}\rangle^2 = c_0^2\mathbf{v}_{\min}^\top \Sigma_{\mathcal{H},h}\mathbf{v}_{\min} = \sigma_{\min}(\Sigma_{\mathcal{H},h})c_0^2.$$

and

$$\mathbb{E}_{\pi_b}\left[(\mathcal{B}^{\mathcal{S}}V)(s_h)^2\right] \geq \min_{s_h} d^{\pi_b}(s_h) \cdot \|c_0\mathbf{v}_{\min}\|_2^2 = \min_{s_h} d^{\pi_b}(s_h) \cdot c_0^2.$$

Therefore

$$\sqrt{\frac{\mathbb{E}_{\pi_b}[(\mathcal{B}^{\mathcal{S}}V)(s_h)^2]}{\mathbb{E}_{\pi_b}[(\mathcal{B}^{\mathcal{H}}V)(\tau_h)^2]}} \geq \sqrt{\frac{\min_{s_h} d^{\pi_b}(s_h) \cdot c_0^2}{\sigma_{\min}(\Sigma_{\mathcal{H},h})c_0^2}} = \sqrt{\frac{\min_{s_h} d^{\pi_b}(s_h)}{\sigma_{\min}(\Sigma_{\mathcal{H},h})}}.$$

$\square$

### E.5 Example 6

It suffices to show

$$\Sigma_{\mathcal{H},h}\mathbf{1} = \mathbb{E}_{\pi_b}[\mathbf{b}(\tau_h)\mathbf{b}(\tau_h)^\top]\mathbf{1} = \mathbb{E}_{\pi_b}[\mathbf{b}(\tau_h)\mathbf{b}(\tau_h)^\top\mathbf{1}] = \mathbb{E}_{\pi_b}[\mathbf{b}(\tau_h)] = \mathbf{b}_h^{\pi_b}.$$

### E.6 Proof of Lemma 8

The key is to notice that $(\mathbf{b}_h^{\pi_e})^\top$ is non-negative, so

$$\begin{aligned}
(\mathbf{b}_h^{\pi_e})^\top \Sigma_{\mathcal{H},h}^{-1}\mathbf{b}_h^{\pi_e} &\leq (\mathbf{b}_h^{\pi_e})^\top |\Sigma_{\mathcal{H},h}^{-1}(\mathbf{b}_h^{\pi_e})| && (|\cdot| \text{ is pointwise absolute value}) \\
&\leq (\mathbf{b}_h^{\pi_e})^\top \mathbf{1} \cdot \|\Sigma_{\mathcal{H},h}^{-1}(\mathbf{b}_h^{\pi_e})\|_\infty && (\text{using the non-negativity of } (\mathbf{b}_h^{\pi_e})^\top \text{ again}) \\
&= \|\Sigma_{\mathcal{H},h}^{-1}(\mathbf{b}_h^{\pi_e})\|_\infty.
\end{aligned}$$

### E.7 Proof of Lemma 5

According to Eq. (5), we use $L_1/L_\infty$ Hölder's inequality and obtain that for any $f_h$

$$
\begin{aligned}
|V_{\mathcal{F}}(f_h)| &\leq \left\| \frac{\mathbf{u}(f_h)}{Z^R(f_h)} \right\|_1 \left\| (\Sigma_{\mathcal{F},h}^R)^{-1} V_{\mathcal{S}}^{\pi_e} \right\|_\infty \\
&= R^+(f_h) \left\| \frac{\mathbf{u}(f_h)}{Z(f_h)} \right\|_1 \left\| (\Sigma_{\mathcal{F},h}^R)^{-1} V_{\mathcal{S}}^{\pi_e} \right\|_\infty \\
&\leq R^+(f_h) C_{\mathcal{F},\infty} \\
&\leq H C_{\mathcal{F},\infty}.
\end{aligned}
$$

The second inequality is from Assumption 9 and $\frac{\mathbf{u}(f_h)}{Z(f_h)}$ is a stochastic vector. The last step is from the boundedness of the reward. $\qquad\square$

### E.8 Example 4

We use $\mathrm{diag}(\cdot)$ both for creating a diagonal matrix with an input vector and for taking the diagonal vector out of a matrix.

$$
\Sigma_{\mathcal{F},h}^R \mathbf{1} = M_{\mathcal{F},h} (Z_h^R)^{-1} M_{\mathcal{F},h}^\top \mathbf{1} = M_{\mathcal{F},h} (Z_h^R)^{-1} \mathrm{diag}(Z_h)^\top = M_{\mathcal{F},h} R_h^+ = V_{\mathcal{S}}^{\pi_e}.
$$

### E.9 Example 9

The calculation is similar to that of Example 2 in Appendix D.3, except that the numerator has an extra $R^+(f_h)$. Therefore, when calculating the $i$-th diagonal entry of $\Sigma_{\mathcal{F},h}^R$, the final sum is calculating the expectation of $R^+(f_h)$ conditioned on $s_h = i$ under policy $\pi_b$, which is the definition of $V_{\mathcal{S}}^{\pi_b}(i)$.

### E.10 Proof of Theorem 7

In the proof of Theorem 2, we have

$$
\sum_{h=1}^H \mathbb{E}_{\pi_b} \left[ (\mathcal{B}^{\mathcal{H}} \widehat{V})(\tau_h)^2 \right] \leq 10 \varepsilon_{\mathrm{stat}}, \quad \varepsilon_{\mathrm{stat}} := \frac{225 H \bar{C}^2 \log(4|\mathcal{V}||\Xi|/\delta)}{n}.
$$

We observe that

$$
\begin{aligned}
\left| \mathbb{E}_{\pi_e} \left[ (\mathcal{B}^{\mathcal{S}} \widehat{V})(s_h) \right] \right| &= \left| \mathbb{E}_{\tau_h \sim d_h^{\pi_e}} \mathbb{E}_{s_h \sim \mathbf{b}(\tau_h)} \left[ (\mathcal{B}^{\mathcal{S}} \widehat{V})(s_h) \right] \right| \\
&= \left| \mathbb{E}_{\pi_e} \left[ (\mathcal{B}^{\mathcal{H}} \widehat{V})(\tau_h) \right] \right| \\
&= \left| \mathbb{E}_{\pi_b} \left[ w^\star(\tau_h) (\mathcal{B}^{\mathcal{H}} \widehat{V})(\tau_h) \right] \right| \\
&\leq \| w^\star \|_{2, d_h^{\pi_b}} \sqrt{ \mathbb{E}_{\pi_b} \left[ (\mathcal{B}^{\mathcal{H}} \widehat{V})(\tau_h)^2 \right] } \\
&\leq \sqrt{ C_{\mathcal{H},2} \mathbb{E}_{\pi_b} \left[ (\mathcal{B}^{\mathcal{H}} \widehat{V})(\tau_h)^2 \right] }
\end{aligned}
$$

The third equality is from Definition 10, the first inequality is from Cauchy-Schwarz inequality and the last inequality is from Lemma 6. Then, we have

$$
\begin{aligned}
\left| \sum_{h=1}^H \mathbb{E}_{\pi_e} \left[ (\mathcal{B}^{\mathcal{S}} \widehat{V})(s_h) \right] \right| &\leq \sqrt{ H \sum_{h=1}^H \left( \mathbb{E}_{\pi_e} \left[ (\mathcal{B}^{\mathcal{S}} \widehat{V})(s_h) \right] \right)^2 } \\
&\leq \sqrt{ 10 H C_{\mathcal{H},2} \varepsilon_{\mathrm{stat}} } \\
&\leq 50 H \bar{C} \sqrt{ \frac{C_{\mathcal{H},2} C_\mu \log(4|\mathcal{V}||\Xi|/\delta)}{n} } \\
&\leq c H^2 (C_{\mathcal{F},\infty} + 1) \sqrt{ \frac{C_{\mathcal{H},2} C_\mu \log(4|\mathcal{V}||\Xi|/\delta)}{n} }.
\end{aligned}
$$

The last step is from Lemma 5, $C_{\mathcal{V}} \leq c\|V_{\mathcal{F}}\|_\infty$ and $C_\Xi \leq c(\|V_{\mathcal{F}}\|_\infty + 1)$. The proof is completed after invoking Lemma 1. $\qquad\square$

## E.11  Proof of Theorem 9

Finally, we prove Theorem 9. The proof uses the similar idea from Xie and Jiang [2020]. For any $V \in \mathcal{V}$ and $w \in \mathcal{W}$, we define the population loss estimator $\mathcal{L}_{d^{\pi_b}}$ and the empirical loss estimator $\mathcal{L}_{\mathcal{D}}$ as follows

$$\mathcal{L}_{d^{\pi_b}}(V, w) := \sum_{h=1}^{H} \mathbb{E}_{\pi_b} \left[ w(\tau_h)(\mathcal{B}^{\mathcal{H}} V)(\tau_h) \right]$$

$$\mathcal{L}_{\mathcal{D}}(V, w) := \sum_{h=1}^{H} \mathbb{E}_{\mathcal{D}} \left[ w(\tau_h) \left( \mu(o_h, a_h) \left( r_h + V(f_{h+1}) \right) - V(f_h) \right) \right].$$

We then invoke Lemma 1 and obtain that

$$\left| J(\pi_e) - \mathbb{E}_{\pi_b}[\widehat{V}(f_1)] \right| = \left| \sum_{h=1}^{H} \mathbb{E}_{s_h \sim d^{\pi_e}} \left[ (\mathcal{B}^{\mathcal{S}} \widehat{V})(s_h) \right] \right|.$$

We observe that

$$\mathbb{E}_{s_h \sim d^{\pi_e}} \left[ (\mathcal{B}^{\mathcal{S}} \widehat{V})(s_h) \right] = \mathbb{E}_{\tau_h \sim d^{\pi_e}} \left[ \mathbb{E}_{s_h \sim \mathbf{b}(\tau_h)} \left[ (\mathcal{B}^{\mathcal{S}} \widehat{V})(s_h) \right] \right]$$

$$= \mathbb{E}_{\tau_h \sim d^{\pi_e}} \left[ (\mathcal{B}^{\mathcal{H}} \widehat{V})(\tau_h) \right]$$

$$= \mathbb{E}_{\pi_b} \left[ w^\star(\tau_h)(\mathcal{B}^{\mathcal{H}} \widehat{V})(\tau_h) \right],$$

where the last step is from Definition 10. Therefore, our goal is to bound $\left| \sum_{h=1}^{H} \mathbb{E}_{\pi_b} \left[ w^\star(\tau_h)(\mathcal{B}^{\mathcal{H}} \widehat{V})(\tau_h) \right] \right| = \left| \mathcal{L}_{d^{\pi_b}}(\widehat{V}, w^\star) \right|$. Let

$$\widehat{w} := \operatorname*{argmin}_{w \in \mathrm{sp}(\mathcal{W})} \max_{V \in \mathcal{V}} \left| \sum_{h=1}^{H} \mathbb{E}_{\pi_b} \left[ (w^\star(\tau_h) - w(\tau_h)) \cdot (\mathcal{B}^{\mathcal{H}} V)(\tau_h) \right] \right|,$$

$$\widetilde{V} := \operatorname*{argmin}_{V \in \mathcal{V}} \sup_{w \in \mathrm{sp}(\mathcal{W})} \left| \sum_{h=1}^{H} \mathbb{E}_{\pi_b} \left[ w(\tau_h) \cdot (\mathcal{B}^{\mathcal{H}} V)(\tau_h) \right] \right|.$$

Then, we subtract the approximation error of $\widehat{w}$ from our objective,

$$\left| \sum_{h=1}^{H} \mathbb{E}_{\pi_b} \left[ w^\star(\tau_h)(\mathcal{B}^{\mathcal{H}} \widehat{V})(\tau_h) \right] \right| = \left| \sum_{h=1}^{H} \mathbb{E}_{\pi_b} \left[ (w^\star(\tau_h) - \widehat{w}(\tau_h)) \cdot (\mathcal{B}^{\mathcal{H}} \widehat{V})(\tau_h) \right] + \sum_{h=1}^{H} \mathbb{E}_{\pi_b} \left[ \widehat{w}(\tau_h) \cdot (\mathcal{B}^{\mathcal{H}} \widehat{V})(\tau_h) \right] \right|$$

$$\leq \left| \sum_{h=1}^{H} \mathbb{E}_{\pi_b} \left[ (w^\star(\tau_h) - \widehat{w}(\tau_h)) \cdot (\mathcal{B}^{\mathcal{H}} \widehat{V})(\tau_h) \right] \right| + \left| \sum_{h=1}^{H} \mathbb{E}_{\pi_b} \left[ \widehat{w}(\tau_h) \cdot (\mathcal{B}^{\mathcal{H}} \widehat{V})(\tau_h) \right] \right|$$

$$\leq \epsilon_{\mathcal{W}} + \left| \sum_{h=1}^{H} \mathbb{E}_{\pi_b} \left[ \widehat{w}(\tau_h) \cdot (\mathcal{B}^{\mathcal{H}} \widehat{V})(\tau_h) \right] \right|.$$

Next, we consider the approximation error of $\widetilde{V}$ and obtain that

$$\left| \sum_{h=1}^{H} \mathbb{E}_{\pi_b} \left[ \widehat{w}(\tau_h) \cdot (\mathcal{B}^{\mathcal{H}} \widehat{V})(\tau_h) \right] \right| = \epsilon_{\mathcal{V}} + \left| \sum_{h=1}^{H} \mathbb{E}_{\pi_b} \left[ \widehat{w}(\tau_h) \cdot (\mathcal{B}^{\mathcal{H}} \widehat{V})(\tau_h) \right] \right| - \sup_{w \in \mathrm{sp}(\mathcal{W})} \left| \sum_{h=1}^{H} \mathbb{E}_{\pi_b} \left[ w(\tau_h) \cdot (\mathcal{B}^{\mathcal{H}} \widetilde{V})(\tau_h) \right] \right|.$$

We then connect $\mathcal{L}_{d^{\pi_b}}(V, w)$ with $\mathcal{L}_{\mathcal{D}}(V, w)$ as follows,

$$
\left| \sum_{h=1}^{H} \mathbb{E}_{\pi_b} \left[ \widehat{w}(\tau_h) \cdot (\mathcal{B}^{\mathcal{H}} \widehat{V})(\tau_h) \right] \right| - \sup_{w \in \mathrm{sp}(\mathcal{W})} \left| \sum_{h=1}^{H} \mathbb{E}_{\pi_b} \left[ w(\tau_h) \cdot (\mathcal{B}^{\mathcal{H}} \widetilde{V})(\tau_h) \right] \right|
$$

$$
\leq \sup_{w \in \mathrm{sp}(\mathcal{W})} \left| \sum_{h=1}^{H} \mathbb{E}_{\pi_b} \left[ w(\tau_h) \cdot (\mathcal{B}^{\mathcal{H}} \widehat{V})(\tau_h) \right] \right| - \sup_{w \in \mathrm{sp}(\mathcal{W})} \left| \sum_{h=1}^{H} \mathbb{E}_{\pi_b} \left[ w(\tau_h) \cdot (\mathcal{B}^{\mathcal{H}} \widetilde{V})(\tau_h) \right] \right|
$$

$$
= \max_{w \in \mathcal{W}} \left| \mathcal{L}_{d^{\pi_b}}(\widehat{V}, w) \right| - \max_{w \in \mathcal{W}} \left| \mathcal{L}_{d^{\pi_b}}(\widetilde{V}, w) \right|
$$

$$
= \max_{w \in \mathcal{W}} \left| \mathcal{L}_{d^{\pi_b}}(\widehat{V}, w) \right| - \max_{w \in \mathcal{W}} \left| \mathcal{L}_{\mathcal{D}}(\widehat{V}, w) \right| + \max_{w \in \mathcal{W}} \left| \mathcal{L}_{\mathcal{D}}(\widehat{V}, w) \right| - \max_{w \in \mathcal{W}} \left| \mathcal{L}_{d^{\pi_b}}(\widetilde{V}, w) \right|
$$

$$
\leq \max_{w \in \mathcal{W}} \left| \mathcal{L}_{d^{\pi_b}}(\widehat{V}, w) \right| - \max_{w \in \mathcal{W}} \left| \mathcal{L}_{\mathcal{D}}(\widehat{V}, w) \right| + \max_{w \in \mathcal{W}} \left| \mathcal{L}_{\mathcal{D}}(\widetilde{V}, w) \right| - \max_{w \in \mathcal{W}} \left| \mathcal{L}_{d^{\pi_b}}(\widetilde{V}, w) \right|
$$

$$
\leq \max_{w \in \mathcal{W}} \left| \mathcal{L}_{d^{\pi_b}}(\widehat{V}, w) - \mathcal{L}_{\mathcal{D}}(\widehat{V}, w) \right| + \max_{w \in \mathcal{W}} \left| \mathcal{L}_{d^{\pi_b}}(\widetilde{V}, w) - \mathcal{L}_{\mathcal{D}}(\widetilde{V}, w) \right|.
$$

The first equality is from that $\sup_{w \in \mathrm{sp}(\mathcal{W})} |f(\cdot)| = \max_{w \in \mathcal{W}} |f(\cdot)|$ for any linear function $f(\cdot)$. The second inequality is from the optimality of $\widehat{V}$.

For any fixed $V, w$, let random variable $X = \sum_{h=1}^{H} w(\tau_h) \left( \mu(o_h, a_h)(r_h + V(f_h)) - V(f_{h+1}) \right)$, $\mathbb{E}_{\pi_b}[X] = \mathcal{L}_{d^{\pi_b}}(V, w)$. Recall that $C_{\mathcal{V}} := \max_{V \in \mathcal{V}} \|V\|_\infty$ and $C_{\mathcal{W}} := \max_h \sup_{w \in \mathcal{W}} \|w\|_\infty$. For $|X|$, we have $|X| \leq 2H C_\mu (1 + C_{\mathcal{V}}) C_{\mathcal{W}}$. For the variance, we have

$$
\mathrm{Var}_{\pi_b}[X]
$$

$$
\leq H \sum_{h=1}^{H} \mathbb{E}_{\pi_b} \left[ w(\tau_h)^2 (\mu(o_h, a_h)(r_h + V(f_{h+1})) - V(f_h))^2 \right]
$$

$$
\leq H \sum_{h=1}^{H} \mathbb{E}_{\pi_b} \left[ w(\tau_h)^2 \left( 2C_{\mathcal{V}}^2 + 2(1 + C_{\mathcal{V}})^2 \mu(o_h, a_h)^2 \right) \right]
$$

$$
\leq 4H^2 (1 + C_{\mathcal{V}})^2 C_\mu C_{\mathcal{W}}^2.
$$

From Bernstein's inequality, with probability at least $1 - \delta$, we have

$$
|\mathcal{L}_{d^{\pi_b}}(V, w) - \mathcal{L}_{\mathcal{D}}(V, w)| \leq 2H(1 + C_{\mathcal{V}}) C_{\mathcal{W}} \sqrt{\frac{C_\mu \log \frac{2}{\delta}}{n}} + \frac{2H(1 + C_{\mathcal{V}}) C_{\mathcal{W}} C_\mu \log \frac{2}{\delta}}{n}.
$$

Taking the union bound and we obtain

$$
\max_{w \in \mathcal{W}} \left| \mathcal{L}_{d^{\pi_b}}(\widehat{V}, w) - \mathcal{L}_{\mathcal{D}}(\widehat{V}, w) \right| + \max_{w \in \mathcal{W}} \left| \mathcal{L}_{d^{\pi_b}}(\widetilde{V}, w) - \mathcal{L}_{\mathcal{D}}(\widetilde{V}, w) \right|
$$

$$
\leq 4H(1 + C_{\mathcal{V}}) C_{\mathcal{W}} \sqrt{\frac{C_\mu \log \frac{2|\mathcal{V}||\mathcal{W}|}{\delta}}{n}} + \frac{4H(1 + C_{\mathcal{V}}) C_{\mathcal{W}} C_\mu \log \frac{2|\mathcal{V}||\mathcal{W}|}{\delta}}{n}.
$$

According to Lemma 5, Assumption 12, $C_{\mathcal{V}} \leq c\|V_{\mathcal{F}}\|_\infty$ and $C_{\mathcal{W}} \leq c\|w^\star\|_\infty$, we further have

$$
\left| J(\pi_e) - \mathbb{E}_{\pi_b}[\widehat{V}(f_1)] \right| \leq \epsilon_{\mathcal{V}} + \epsilon_{\mathcal{W}} + cH^2 C_{\mathcal{H},\infty}(C_{\mathcal{F},\infty} + 1) \sqrt{\frac{C_\mu \log \frac{2|\mathcal{V}||\mathcal{W}|}{\delta}}{n}} + \frac{cH^2 C_{\mathcal{H},\infty}(C_{\mathcal{F},\infty} + 1) C_\mu \log \frac{2|\mathcal{V}||\mathcal{W}|}{\delta}}{n}.
$$

The proof is completed by using Hoeffding's inequality to bound $|\mathbb{E}_{\mathcal{D}}[\widehat{V}(f_1)] - \mathbb{E}_{\pi_b}[\widehat{V}(f_1)]|$. $\qquad\square$

