# OpenReview forum: "On the Curses of Future and History in Future-dependent Value Functions for Off-policy Evaluation"
_NeurIPS.cc/2024/Conference — NeurIPS 2024 poster_

### Official Review · Reviewer_NgTY · 2024-06-14

**Soundness:** 3
**Presentation:** 4
**Contribution:** 3
**Rating:** 7
**Confidence:** 3

**Summary:**

This paper proposes new off-policy estimators for POMDPs without exponential variance with the horizon. Specifically, outcome coverage and belief coverage are assumed which capture the past and future information, respectively. This framework reduces the estimation guarantees from exponential to polynomial.

**Strengths:**

1. This paper gives comprehensive analysis in theory, which makes the proposed paradigm sound.

2. The improvement from exponential to polynomial is impressive.

**Weaknesses:**

1. The new algorithm in Section 5.3 has not been studied thoroughly with experiments.

**Questions:**

1. Is it possible to extend the analysis to infinite horizon?

2. Can you justify the full row rank assumption in Assumption 2?

**Limitations:**

The authors addressed the limitations in Section 6.

---

> ### Author Rebuttal · Authors · 2024-08-02
>
> We thank the reviewer for their appreciation of our work and the valuable comments.
>
> > **”Is it possible to extend the analysis to infinite horizon?”**
>
> We believe the answer is yes. In fact, the work of Uehara et al. [2022a] (which we build on) is in the infinite-horizon discounted setting. However, as we discussed in Appendix C.2 (page 17), working with POMDPs and performing the kind of analyses we do in the infinite-horizon setting is really messy. (Just to give a taste: in the infinite-horizon setting, computing belief state requires us to trace back history indefinitely, and in the finite-horizon setting we only need to trace back to the beginning of the episode; see more detailed discussion in C.2) In contrast, the finite-horizon formulation is much cleaner and we are able to re-express the ideas and analyses in Uehara et al. [2022a] in much more elegant forms, which is significant given the complexity of the POMDP formulation (i.e., we want to simplify all aspects as much as we can, as long as it does not lose the essence of the setting). If one is willing to put up with the messiness of the infinite-horizon setting, we believe it should be possible to translate our results into the infinite-horizon setting that Uehara et al. [2022a] took.
>
> ---
>
> > **“Can you justify the full row rank assumption in Assumption 2?"**
>
> These assumptions are made for technical convenience, as mentioned in Line 108. Our main results still hold after some modifications even if they do not hold. Taking L2 belief coverage (Assumption 11) as an example: if $\Sigma\_{H, h}$ is not invertible, but $b\_h^{\pi\_e}$ lies in the subspace of $\Sigma\_{H, h}$, we can still define L2 belief coverage by replacing inverse with pseudo-inverse, and the rest of the analyses and the main results still hold. (If $b\_h^{\pi\_e}$ does not lie in the subspace, we can just define the coverage parameter to be infinite, i.e., no guarantee can be given.) Assuming full-rankness is just a convenient way to avoid dealing with these hassles.
>
> To recap, what really matters is whether (and to what extent)  $b\_h^{\pi\_e}$ lies in the subspace of $\Sigma\_{H, h}$; the binary notion of full row rank or not of $\Sigma\_{H, h}$ does not really matter and is made only for simplifying presentations. Moreover, these full-rank assumptions are intimately connected to the notion of “core histories” and “core tests (future events)” in the PSR literature, which the future-dependent value function framework draws inspiration from (see their connection in Uehara et al. [2022a]).

---

> > ### Comment · Reviewer_NgTY · 2024-08-12
> >
> > Thank the authors for the response. I am maintaining my score.

---

### Official Review · Reviewer_jTik · 2024-07-02

**Soundness:** 4
**Presentation:** 3
**Contribution:** 4
**Rating:** 7
**Confidence:** 3

**Summary:**

This paper addresses off-policy evaluation in the context of POMDP, aiming to develop estimators that avoid exponential dependence on the horizon. Paper introduces two novel coverage assumptions --- outcome coverage and belief coverage --- tailored to POMDPs to achieve polynomial bounds on estimation guarantees. Specifically, outcome coverage ensures boundedness of future-dependent value functions (FDVF) which targets the shift from $\pi_b\rightarrow\pi_e$, while belief coverage deals with IV ($\mathcal{B}^\mathcal{H}\rightarrow\mathcal{B}^\mathcal{S}$) and Dr ($\pi_b\rightarrow\pi_e$) jointly. The work leverages unique properties of POMDP coverage conditions, avoiding explicit dependence on the latent state space size. The MIS algorithm is proposed to provide interpretations concerning the sample complexity under theses new assumptions.

**Strengths:**

- The paper is well-structured and exceptionally clear, making it easy to read. The problem formulation, assumptions and subsequent results are presented with great clarity. The mathematical derivations appear sound.
- The new coverage assumptions are novel and effectively solve the non-trivial problems (1) the counterpart of bounded density ratio, the widely adopted coverage assumption for offline MDP, in the context of POMDPs (2) avoid exponentials.
- A information-theoretical algorithm, MIS, is provided for interpretation of sample complexity

**Weaknesses:**

Minor:  I appreciate this work as a theoretical contribution. It would be even more promising if some preliminary experiments were conducted, as OPE using such a history weight function is a novel idea and its empirical performance cannot be anticipated.

**Questions:**

- As pointed out in the title, FDVF is the key tool to deal with OPE for POMDPs. I wonder if the authors could comment more on why such "future-dependence" is necessary to deal with partial observation?
- From my intepretation, it seems that $L_\infty$ assumptions should be looser ones and can better accomodates the PO. Why are the $L_2$ assumptions emphasized, or do they offer any unnoticed benefits?
- Minor: in line 312, should it be $d^{\pi_e}$?

**Limitations:**

The proposed method is limited to handling memoryless policies and history-dependent policies with structured constraints, but this limitation does not hurt the paper's contributions considering the hardness explained in the appendix.

---

> ### Author Rebuttal · Authors · 2024-08-02
>
> We thank the reviewer for their appreciation of our work and the valuable comments.
>
> > **”Why such `future-dependence’ is necessary to deal with partial observation?”**
>
> As we mentioned in the paper, the most obvious thing to try is to use history-dependent value functions by converting the POMDP into a history-based MDP. However, a straightforward extension incurs the curse of horizon, which shows that some new ideas are needed to overcome this difficulty. We did not claim that “future-dependence” is absolutely “necessary”,
> just that it is a promising idea for addressing this issue. Future-dependence may not be the only idea that works, and other plausible approaches remain to be explored in future works.
>
> ---
>
> > **”From my interpretation, it seems that $L\_\infty$ assumptions should be looser ones and can better accommodate the PO. Why are the t $L\_2$ assumptions emphasized, or do they offer any unnoticed benefits?”**
>
> As the reviewer also noticed, $L\_\infty$ assumptions are looser than their $L\_2$ counterparts (but the $L\_2$ versions often require additional assumptions, e.g., Assumption 8). So whenever no additional assumption is needed, using $L\_2$ assumptions lead to tighter bounds and better guarantees (e.g., Theorem 7 depends on the $L\_2$ version of belief coverage, and no additional regularity assumption is needed).
>
> Another reason we emphasized $L\_2$ assumptions is its familiarity to the RL theory audience, that they look very similar to coverage coefficients in the linear MDP literature (Line 246). On the other hand, $L\_\infty$ assumptions are very … alien, which can look problematic at the first glance (Line 341) and its validity largely relies on the $L\_1$ normalization of belief vectors, a property rarely found in other settings in RL theory. Therefore, we start with $L\_2$ assumptions for a more gentle introduction, and describe $L\_\infty$ later as an improvement.
>
> ---
>
> > Line 312
>
> It is indeed a typo; thanks for pointing it out!

---

> > ### Comment · Reviewer_jTik · 2024-08-08
> >
> > I thank the authors for the detailed response. I am maintaining my original score and remain in favor of acceptance.

---

### Official Review · Reviewer_zNfE · 2024-07-12

**Soundness:** 3
**Presentation:** 3
**Contribution:** 2
**Rating:** 6
**Confidence:** 3

**Summary:**

This paper studied the finite sample guarantee of future-dependent value function (FDVF) based method for policy evaluation problem in POMDPs. The existing guarantee depends on the boundedness of the FDVF which can be exponential in horizon. The authors studied this quantity and proposed new coverage assumptions with intuitive explanation under which FDVF can be well-bounded achieving polynomial guarantee on model parameters. Besides, they also quantified a conversion ratio between Bellman residual on states and history.

**Strengths:**

1. The paper clearly pointed out the problem of the existing sample guarantee (Theorem 2) along with illustrating examples, such as the boundedness of $C_\Xi$ can be exponential in horizon.
2. The assumptions proposed are intuitively interpretable and successfully solve the problem.

**Weaknesses:**

1. For the boundedness of FDVF, there isn't a clear discussion on the strictness of the assumptions in all cases except in some intuitive examples. A claim that $C_{\mathcal{F}, V}$ is polynomially bounded in all cases is needed.
2. To claim a fully polynomial guarantee, the paper didn't discuss on $C_\Xi$, though it also appears in the sample guarantee and can dominate $C_{\mathcal{V}}+1$.

**Questions:**

See weaknesses.

**Limitations:**

See weaknesses.

---

> ### Author Rebuttal · Authors · 2024-08-02
>
> We thank the reviewer for their valuable comments.
>
> > **“For the boundedness of FDVF, there isn't a clear discussion on the strictness of the assumptions in all cases except in some intuitive examples. A claim that $C\_{F, V}$ is polynomially bounded in all cases is needed.”**
>
> $C\_{F, V}$ is the (L2) outcome coverage. As its name suggests, it is a _coverage_ parameter, which describes the extent to which the data (sampled from the behavior policy) contains information about the target policy. Coverage parameters will **not** be bounded in all cases. For example, in the MDP literature, (the boundedness of) the state-density ratios are a standard form of coverage parameter, which can be easily infinite with a poor offline data distribution. Nevertheless, we view state-density ratio as a “polynomial quantity” because there are natural settings where the cumulative importance weights are exponential yet the state-density ratio remain small (see e.g., the example on page 3 of Liu et al. [2018]: “Breaking the Curse of Horizon: Infinite-Horizon Off-Policy Estimation”). Indeed, the wide acceptance of state-density ratio as an appropriate coverage parameter is precisely established by studying “intuitive examples”, just as we did in this paper.
>
> Note that this is a major difference between MDPs and POMDPs (and not realizing the difference might be why the reviewer made this comment in the first place): in MDPs, value functions are always bounded and they have nothing to do with the offline data distribution. In the future-dependent value function (FDVF) framework, however, these FDVFs are properties of both the behavior and the target policies (Line 131), and their existence and boundedness depend on a new form of coverage not seen in MDPs (Line 248), namely the outcome coverage.
>
>
> ---
>
> > **“the paper didn't discuss on $C\_{\Xi}$”**
>
> $C\_{\Xi}$ is assumed to be bounded by  $c (\\\|V\_F\\\|\_\infty+1)$ (Line 327), with $c$ being an absolute constant. The rationale is exactly the same as for $C\_V$ (see below), so we omitted the explanation for $C\_{\Xi}$ due to space limit at submission time. That said, the reviewer is right that we should have discussed this explicitly, and we will add in revision.
>
> Recall that for $C\_V$, we wrote in Line 322:
>
> > To highlight the dependence of $\\\|V\_F\\\|\_\infty$ on the proposed coverage assumptions, we follow Xie and Jiang [2020] to assume that **the range of the function classes is not much larger than that of the function it needs to capture** [i.e., $C\_V \le c \\\|V\_F\\\|\_\infty$ for absolute constant c, as in Line 327].
>
> We made a similar assumption for $C\_{\Xi}$ in Line 327 for exactly the same reason: the functions that $\Xi$ needs to capture are $\\\{B^H V: V \in \mathcal{V}\\\}$ (the Bellman completeness in Assumption 6, Line 152). From the definition of $B^H V$ (Eq.(2)), we know that its boundedness is immediately provided by the boundedness of $V \in \mathcal{V}$ (i.e., $2C\_V + 1$). The final assumption on $C\_{\Xi}$ follows from combining this with the bound on $C\_V$ above.

---

> > ### Comment · Reviewer_zNfE · 2024-08-12
> >
> > I appreciate the authors' efforts on the rebuttal. The elaboration sounds reasonable to me and I would like to increase my score to 6.

---

### Official Review · Reviewer_Djpv · 2024-07-15

**Soundness:** 2
**Presentation:** 2
**Contribution:** 2
**Rating:** 5
**Confidence:** 1

**Summary:**

This paper studies off-policy evaluation in POMDPs and introduces two novel coverage concepts: outcome coverage and belief coverage. Outcome coverage uses a weighted norm to ignore unimportant futures in future-dependent value functions. Belief coverage is related to the covariance matrix of belief states under the behavior policy. The paper argues that these conditions are sufficient for the reward of the evaluation policy to be close to the estimated value function for the dataset generated by the behavior policy.

**Strengths:**

To be honest, this paper is quite far beyond my abilities with RL theory. I was able to identify two positive aspects.

1. The paper operates in the POMDP setting, and (as far as I can tell) does not assume the learning agent ever has access to the set of latent states S. If I'm incorrect about this, I ask the authors to let me know so I can fix my understanding.

2. The paper starts by suggesting that we will be looking for solutions in $mathcal{F}_h$, but points out correctly that this will give us exponential dimensionality, and subsequently finds another approach with better properties.

**Weaknesses:**

The paper's main weakness is that it targets a very limited audience of theoreticians only. There is extremely little text providing intuition or any sort of motivation for the research questions, approach or solution.

Despite significant familiarity with off-policy evaluation, importance sampling, POMDPs, etc., I was unfortunately unable to:
- follow most of the paper,
- understand the main questions being asked,
- see what this would be useful for,
- evaluate the reasonableness of the assumptions,
- check the proofs,
- appreciate the consequences of the results.

In short, if the paper is intended to be read by anyone outside of a very limited audience, it will need to be substantially rewritten so as to be much more accessible. I'm not even sure if my summary is correct.

**Questions:**

Can the authors please provide a summary of the work that a non-theoretician might understand and appreciate if they have a strong background in the empirical side of the relevant concepts?

---

> ### Author Rebuttal · Authors · 2024-08-02
>
> We thank the reviewer for their valuable comments.
>
> > **“the paper … does not assume the learning agent ever has access to the set of latent states S. If I'm incorrect about this, I ask the authors to let me know ...”**
>
> You are right. As in the standard POMDP setting, the latent states $s\_h$ are not included in the dataset received by the agent performing off-policy evaluation. This has also been made explicit in Lines 66 and 67. Both the behavior and the target policies also only operate on the observables, i.e., they cannot depend on the latent states (see Footnote 1).
>
> ---
>
> > **“The paper's main weakness is that it targets a very limited audience of theoreticians. There is extremely little text providing intuition or any sort of motivation for the research questions, approach or solution.”**
>
> We appreciate the comment about readability and will try our best to address it. But yes, this is a pure theory paper mainly aiming at the RL theory audience, a community with a strong presence at NeurIPS. We believe that the study of POMDPs, especially providing results in modern (offline) RL theory frameworks (i.e., providing sample complexity guarantees in terms of proper coverage parameters), is an important and understudied research direction. For MDPs, the research on understanding coverage assumptions (starting from early works of Munos and Szepesvari between 2000 and 2010) has led to growth of the offline RL theory community, and eventually practical offline algorithms that are both theoretically sound and empirically effective (see e.g., ICML 2022 Outstanding Paper Runner-up: “Adversarially trained actor critic for offline reinforcement learning”). What our paper does can be viewed as similar efforts that lay down the theoretical foundations for POMDPs, which may inspire practical algorithms later. We have articulated such a motivation in the context of (offline) RL theory research in the abstract, the introduction, and Section 3; see also our response below to your question on “summary for non-theoretician”. In the revised version, we will include additional text to further clarify the significance and rationale behind our research questions and proposed solutions.
>
> Furthermore, the POMDP theory literature have generally been known to be mathematically involved (e.g., the PSR literature, which the future-dependent value function framework draws inspiration from, has always been a niche yet important topic in RL), and distilling knowledge into an easy-to-understand form often takes multiple papers and years of efforts in such research directions. We believe we have already made progress on this: in Section 3, we rewrote and presented the work of Uehara et al. [2022a] in much simpler forms compared to the original paper, forming a clean basis for our later investigation as well as future work in this direction.
>
> ---
>
> > **“Can the authors please provide a summary of the work that a non-theoretician might understand and appreciate if they have a strong background in the empirical side of the relevant concepts?”**
>
> In OPE in MDPs, methods that learn value functions have the theoretical advantage of paying state density ratios as its coverage parameter, compared to importance sampling that incurs exponential dependence. However, naive extensions and analyses to POMDPs erase such advantages. We identify novel coverage definitions and further develop the future-dependent value function (FDVF) framework that reinstantiate such advantages in POMDPs.

---

> ### Comment · Reviewer_Djpv · 2024-08-09
>
> Thanks for the response. I perhaps spoke too strongly about what I called the very limited audience. Let me amend my statement to simply say that this paper does not feel accessible to me, despite the fact that I am a practitioner with experience in many of the areas discussed here, and with plenty of theory knowledge as well (just not this kind of theory). This feels like a disadvantage for the paper, and it's unfortunately the only aspect I can comment on. There are theory papers I've read that don't feel this way, and I would encourage the authors to try their best to make this work as accessible as possible.

---

### Decision · Program_Chairs · 2024-09-25

**Decision:**

Accept (poster)

**Comment:**

This paper presents innovative coverage assumptions for Off-Policy Evaluation (OPE) in the Partially Observable Markov Decision Process (POMDP) setting. Unlike the naive coverage coefficient definition, these new coefficients do not exhibit a direct exponential dependence on the POMDP's horizon. Building upon these quantities, the authors develop sample-efficient OPE algorithms tailored to this setting.

Currently, the proposed algorithms are only sample efficient and their practical usefulness seems currently limited. Nevertheless, this work is a significant advancement in the OPE for POMDPs setting;  a challenging and highly relevant problem in real-world applications. The proposed approach has the potential to facilitate the design of practical OPE algorithms for POMDPs and offer researchers a fresh perspective on addressing existing challenges in POMDPs. Therefore, I am pleased to support the acceptance of this work.